# Control of osteocyte dendrite formation by Sp7 and its target gene osteocrin

Jialiang S. Wang[1], Tushar Kamath[2,3], Courtney M. Mazur [1], Fatemeh Mirzamohammadi[1,4], Daniel Rotter[1,5], Hironori Hojo[6], Christian D. Castro[1], Nicha Tokavanich[1], Rushi Patel[1], Nicolas Govea[1,7], Tetsuya Enishi [1,8], Yunshu Wu[1,9], Janaina da Silva Martins [1], Michael Bruce[1], Daniel J. Brooks[1,10], Mary L. Bouxsein [1,10], Danielle Tokarz [11,12], Charles P. Lin[11], Abdul Abdul[2,3], Evan Z. Macosko [2,3,13], Melissa Fiscaletti[14], Craig F. Munns[15,16], Pearl Ryder[2,17], Maria Kost-Alimova[2,18], Patrick Byrne[2,18], Beth Cimini [2,17], Makoto Fujiwara[19], Henry M. Kronenberg[1] & Marc N. Wein [1,2,20✉]

Some osteoblasts embed within bone matrix, change shape, and become dendrite-bearing osteocytes. The circuitry that drives dendrite formation during "osteocytogenesis" is poorly understood. Here we show that deletion of *Sp7* in osteoblasts and osteocytes causes defects in osteocyte dendrites. Profiling of Sp7 target genes and binding sites reveals unexpected repurposing of this transcription factor to drive dendrite formation. *Osteocrin* is a Sp7 target gene that promotes osteocyte dendrite formation and rescues defects in Sp7-deficient mice. Single-cell RNA-sequencing demonstrates defects in osteocyte maturation in the absence of Sp7. Sp7-dependent osteocyte gene networks are associated with human skeletal diseases. Moreover, humans with a *SP7R316C* mutation show defective osteocyte morphology. Sp7-dependent genes that mark osteocytes are enriched in neurons, highlighting shared features between osteocytic and neuronal connectivity. These findings reveal a role for Sp7 and its target gene *Osteocrin* in osteocytogenesis, revealing that pathways that control osteocyte development influence human bone diseases.

[1] Endocrine Unit, Massachusetts General Hospital, Harvard Medical School, Boston, MA, USA. [2] Broad Institute of Harvard and MIT, Cambridge, MA, USA. [3] Stanley Center for Psychiatric Research, Broad Institute of Harvard and MIT, Cambridge, MA, USA. [4] Department of Plastic and Reconstructive Surgery, Wright State University, Dayton, OH, USA. [5] University of Applied Sciences Technikum Wien, Vienna, Austria. [6] Center for Disease Biology and Integrative Medicine, The University of Tokyo Graduate School of Medicine, 7-3-1 Hongo, Bunkyo-ku, Tokyo 113-8656, Japan. [7] Department of Anesthesiology, Weill Cornell Medical School, New York, NY, USA. [8] Department of Orthopedic Surgery, Tokushima Municipal Hospital, Tokushima, Japan. [9] State Key Laboratory of Oral Diseases, National Clinical Research Center for Oral Diseases, West China Hospital of Stomatology, Sichuan University, Chengdu, China. [10] Center for Advanced Orthopedic Studies, Department of Orthopedic Surgery, Beth Israel Deaconess Medical Center, Harvard Medical School, Boston, MaA, USA. [11] Advanced Microscopy Program, Center for Systems Biology and Wellman Center for Photomedicine, Massachusetts General Hospital, Harvard Medical School, Boston, MA, USA. [12] Department of Chemistry, Saint Mary's University, Halifax, Canada. [13] Department of Psychiatry, Massachusetts General Hospital, Harvard Medical School, Boston, MA, USA. [14] Pediatric Department, Sainte-Justine University Hospital Centre, Montreal, Canada. [15] Institute of Endocrinology and Diabetes, The Children's Hospital at Westmead, Sydney, NSW, Australia. [16] Discipline of Paediatrics & Child Health, University of Sydney, Sydney 2006, Australia. [17] Broad Institute of Harvard and MIT, Imaging Platform, Cambridge, MA, USA. [18] Broad Institute of Harvard and MIT, Center for the Development of Therapeutics, Cambridge, MA, USA. [19] Department of Pediatrics, Osaka University Graduate School of Medicine, Osaka, Japan. [20] Harvard Stem Cell Institute, Cambridge, MA, USA. ✉email: mnwein@mgh.harvard.edu

The major cell types that govern bone homeostasis are osteoblasts, osteoclasts, and osteocytes. Although the roles of osteoblasts and osteoclasts in bone formation and resorption have been well studied[1,2], those of osteocytes, the most abundant cell type in bone, had been overlooked due to technological limitations and the cells' relatively inaccessible location within the mineralized bone matrix. Recently, emerging evidence has highlighted key roles for osteocytes in bone remodeling[3]. These cells translate external cues, such as hormonal variations and mechanical stresses, into changes in bone remodeling by secreting paracrine-acting factors that regulate osteoblast and osteoclast activity[4]. Furthermore, osteocytes have a unique morphology as they bear multiple long, neuron-like dendritic processes projecting through the lacunar canalicular system in bone[5]. The osteocyte dendritic network confers mechano-sensitivity to these cells and allows for extensive communication among osteocytes and adjacent cells on bone surfaces[6]. Defects in the osteocyte dendrite network may cause skeletal fragility in the setting of aging and glucocorticoid treatment[7,8]. Recent estimates suggest that the osteocyte connectivity network in human bone exhibits the same order of complexity as the network of connections between neurons in the brain[9].

Lineage-specifying transcription factors have been identified for other key cell types in bone;[10] however, lineage-defining transcription factors that coordinate genetic programs associated with osteocyte maturation remain unknown. The goal of this study was to define key gene regulatory circuits that drive osteocyte differentiation and dendrite formation. Osterix/Sp7 (encoded by the *Sp7* gene) is a zinc finger-containing transcription factor essential for osteoblast differentiation and bone formation downstream of Runx2[11]. Common human *SP7*-associated variants are linked to bone mineral density variation and fracture risk[12,13], and rare *SP7* mutations cause recessive forms of osteogenesis imperfecta[14,15]. However, the function of Sp7 in osteocyte maturation remains unknown.

In this work, we delete Sp7 in mature osteoblasts and osteocytes and observe a dramatic skeletal phenotype including cortical porosity, increased osteocyte apoptosis, and severe defects in osteocyte dendrites. Unbiased profiling of Sp7 target genes and binding sites in osteocytes reveals a stage-specific role for this transcription factor during osteocytogenesis. Osteocyte-specific Sp7 target genes have DNA-binding sites distinct from those in osteoblasts and are highly enriched in genes expressed in neurons, highlighting shared molecular links between intercellular communication networks in brain and bone. Amongst osteocyte-specific Sp7 target genes, we identify osteocrin as a secreted factor that promotes osteocyte dendrite formation/maintenance in vitro and in vivo. Single-cell RNA-sequencing identifies discrete populations of cells undergoing the osteoblast-to-osteocyte transition, and dramatic defects in this normal process in the absence of Sp7. Finally, we report overt defects in osteocyte dendrites in humans with bone fragility due to the rare *SP7^{R316C}* mutation. Taken together, these findings highlight a central role for Sp7 in driving cell morphogenesis during osteocytogenesis.

## Results

**Sp7 deletion in mature osteoblasts leads to severe osteocyte dendrite defects.** We sought to identify gene regulatory networks that control osteocytogenesis[16,17]. *Sost* is uniquely expressed in osteocytes but not osteoblasts[18,19], so we hypothesized that transcription factors that drive *Sost* expression will control other aspects of osteocyte function. Combined overexpression of *Atf3*, *Klf4*, *Pax4*, and *Sp7* induced ectopic *Sost* expression in dermal fibroblasts[20]. Therefore, we asked if these transcription factors might participate in eutopic *Sost* expression in Ocy454 cells[21].

Of these factors, only *Sp7* controlled endogenous sclerostin secretion (Supplementary Fig. 1). To examine the role of Sp7 in osteocytes in vivo, we deleted *Sp7* using *Dmp1-Cre*[22], which targets mature osteoblasts and osteocytes (Fig. 1a, e). *Sp7^{OcyKO}* mice showed increased cortical porosity (Fig. 1b, f), reduced cortical bone mineral density (Fig. 1g; Supplementary Data 1a shows cortical bone and trabecular micro-CT parameters for control and *Sp7^{OcyKO}* mice of both sexes), and abnormal intracortical bone remodeling as characterized by the increased intracortical bone formation and resorption (Fig. 1c–d) associated with increased expression of the key osteoclastogenic cytokine *Rankl* (encoded by the *Tnfsf11* gene) (Fig. 1h). Supplemental Data 1b shows static and dynamic histomorphometry measurements performed on trabecular bone surfaces in the distal tibia metaphysis.

Given the dramatic cortical bone phenotypes present in *Sp7^{OcyKO}* mice, we subsequently focused analysis on that skeletal compartment. In situ phalloidin staining of actin filaments and in vivo third-harmonic generation (THG) microscopy[23] both revealed dramatic reductions in osteocyte dendrites and inter-osteocyte connectivity in *Sp7* mutants compared with controls (Fig. 1i, j, m–o; Supplementary Fig. 2[24]). Osteocyte morphology defects seen here are similar to those previously reported in global, inducible *Sp7* mutant mice[25]. Osteocyte apoptosis was elevated in *Sp7^{OcyKO}* mice as assessed by terminal deoxynucleotidyl transferase dUTP nick end labeling (TUNEL) staining (Fig. 1k, q) and immunostaining for activated caspase-3 (Fig. 1l, v), and *Sp7* conditional mutants showed increased numbers of empty lacunae (Fig. 1l, u) without changes in overall lacunar density (Fig. 1t). Dendrite numbers were reduced even in non-apoptotic *Sp7^{OcyKO}* osteocytes (Fig. 1r, v), suggesting that apoptosis may not be the cause of decreased dendrite number. These results indicate that Sp7, a transcription factor crucial for early steps in osteoblast lineage commitment[11], continues to play a key role in maintaining skeletal integrity throughout the osteoblast/osteocyte lineage where it is required for normal dendritic morphology. Without Sp7, dysmorphic osteocytes undergo apoptosis, which triggers *Rankl* upregulation, intracortical remodeling[26,27], and gross defects in cortical bone integrity in long bones.

**Sp7 deficiency in vitro impedes dendrite formation.** Although *Sp7* knockdown caused no obvious morphology differences in MC3T3-E1 (osteoblast-like) normal tissue culture conditions (Fig. 2a), upon growth in 3D collagen gels, cells lacking Sp7 (Supplemental Fig. 1c) showed reduced dendrite numbers compared with control cells (Fig. 2a, b). Similar results were seen in Ocy454 cells, where Sp7 deficiency also caused reduced dendrite development and phalloidin staining in 2D culture (Fig. 2c, Supplemental Fig. 3a). Transcriptomic analysis of 2D- versus 3D-cultured cells revealed coordinate changes in 3D culture consistent with osteocyte maturation (Supplemental Fig. 4a and Supplemental Data 2) including marked upregulation of osteocyte marker genes such as *Dmp1*, *Phex*, and *Ank*. Next, we assessed the functional consequences of *Sp7* knockdown in these cell culture models. Sp7 reduction in Ocy454 cells caused reduced cell numbers over time in culture (Fig. 2d), a defect that is accompanied by unchanged proliferation (Fig. 2e) and, consistent with our in vivo findings (Fig. 1k, l, p, u), increased apoptosis (Fig. 2e). These findings support the utility of 3D culture system for studies on the role of Sp7 in osteocytogenesis, and demonstrate that Sp7 is necessary for osteoblasts to undergo an osteocyte-like morphology change.

**Osteocyte Sp7 target genes are involved in dendrite formation.** The osteocyte network in bone bears similarity to the network of intercellular connections between neurons[9]. In addition,

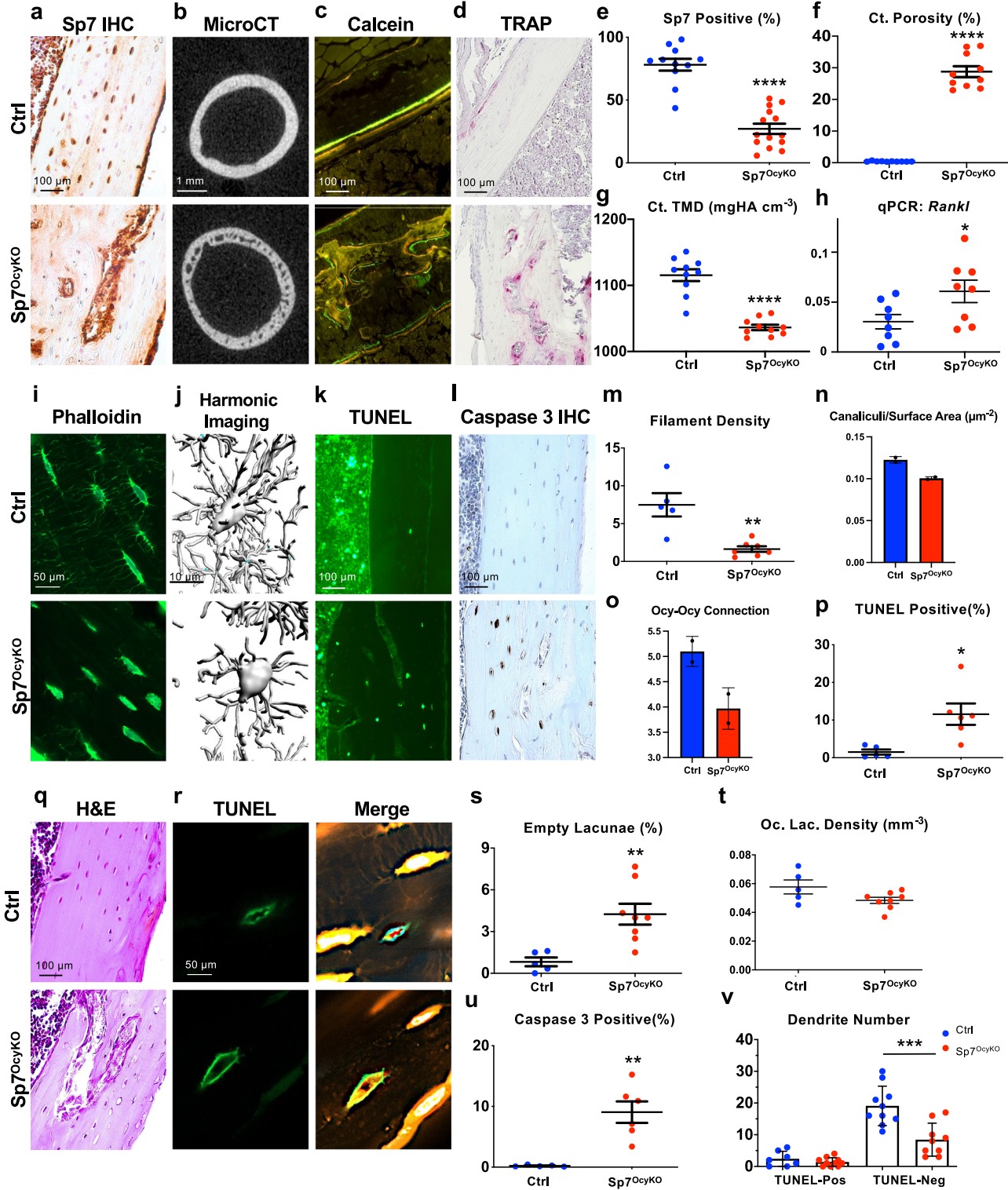

osteocytes have been reported to express certain neuronal transcripts;[28,29] however, the molecular mechanisms used by osteocytes to acquire this gene expression program are unknown. RNA-seq in Sp7-deficient and overexpressing Ocy454 cells (Fig. 3a–b; Supplementary Fig. 4b; Supplementary Data 2) revealed that Sp7-dependent genes are enriched in gene ontology (GO) terms linked to cell projection organization and neuronal development. Merging data sets from Sp7 knockdown and overexpression transcriptomic studies identified a core group of transcripts that were counter-regulated by both genetic

perturbations (Fig. 3c–e). A heatmap representation of differentially expressed genes subdivided by GO terms of interest further highlights core groups of neuronally related transcripts whose expression is regulated by Sp7 in osteocytes (Fig. 3f).

Next, we performed Sp7 ChIP-seq in Ocy454 cells (Fig. 4a; Supplementary Data 3; studies were performed in conditionally-immortalized Ocy454 cells owing to challenges obtaining sufficient numbers of primary osteocytes for ChIP-seq) and compared this data set with the published Sp7 cistromic data in primary osteoblasts (POB)[30] to define osteocyte-specific

**Fig. 1 Severe skeletal defects in $Sp7^{OcyKO}$ mice. a** Sp7 immunohistochemistry was performed on tibiae from 8-week-old control (*Dmp1-Cre; Sp7^{+/+}*) and $Sp7^{OcyKO}$ (*Dmp1-Cre; Sp7^{f/f}*) mice, quantification of Sp7-positive osteocytes is shown in **e. b** Cross-sectional µ-CT images from the femoral midshaft diaphysis reveal increased cortical porosity and reduced mineralization in $Sp7^{OcyKO}$ mice, quantified in **f** and **g. c** 8-week-old mice were labeled with calcein (green) and demeclocycline (red) 7 and 2 days prior to sacrifice, respectively. Non-decalcified sections from the cortical bone in the tibia were analyzed. Control mice show orderly endosteal bone formation. In contrast, $Sp7^{OcyKO}$ animals show abnormal intracortical bone formation. **d** TRAP-stained (red) paraffin-embedded sections from the tibia show an increased number of intracortical osteoclasts $Sp7^{OcyKO}$ in mice. **h** Cortical bone RANKL expression was measured by RT-qPCR. **i** Cryosections from 8-week-old control (*Dmp1-Cre; Sp7^{+/+}*) and $Sp7^{OcyKO}$ (*Dmp1-Cre; Sp7^{f/f}*) tibia were stained with phalloidin to visualize actin filaments, **m** shows quantification indicating reduced filament density (percent of acellular bone matrix occupied by phalloidin-positive filaments) in $Sp7^{OcyKO}$ mice. **j** In vivo third-generation harmonic imaging of the skull was performed to visualize osteocyte cell bodies and canaliculi in the skull. Quantification of defects in canaliculi/surface area and osteocyte–osteocyte connection is shown in **n–o. k–l** Apoptosis in situ was analyzed on tibia sections by TUNEL staining and activated caspase-3 immunohistochemistry; both methods demonstrate increased osteocyte apoptosis in $Sp7^{OcyKO}$ mice versus controls. **p, u** show the quantification. **q** Hematoxylin and eosin-stained paraffin-embedded sections from the tibia show abnormal cortical porosity and empty osteocyte lacunae, quantified in **s–t. r, v** TUNEL staining (green) was performed on control (*Dmp1-Cre; Sp7^{+/+}, Ai14*) and $Sp7^{OcyKO}$ (*Dmp1-Cre; Sp7^{f/f}, Ai14*) mice. Thereafter, osteocyte filaments were visualized by tdTomato fluorescence within dendritic projections. Dendrite number was counted in TUNEL-positive (rare in control mice) and TUNEL-negative osteocytes. Reduced dendrite numbers are noted in non-apoptotic (TUNEL-negative) osteocytes lacking Sp7. *$*p < 0.05$, **$p < 0.01$. ***$p < 0.001$, ****$p < 0.0001$. In graphs in **e–h**, **m–p**, and **s–v**, each data point represents a biologically independent animal. Data are presented as mean values ±SEM. Two-sided student's $t$ tests were used without adjustment for multiple comparisons for all panels except **u** where two-way ANOVA was performed followed by Tukey's multiple comparisons test. Exact $p$ values for comparisons are as follows: **e** <0.0001, **f** <0.0001, **g** <0.0001, **h** 0.0384, **m** 0.0015, **p** 0.0118, **s** 0.0055, **u** 0.0014, **v** 0.0002. Formal statistical analysis was not performed for **n**, **o** where two mice per genotype were analyzed.

(Ocy-specific), primary osteoblast-specific (POB-specific), and shared Sp7 (Ocy-POB) enhancer binding sites (Fig. 4a–b). Ocy-specific Sp7 enhancer peaks are enriched in genes associated with axon guidance (Fig. 4c, left) and Ocy-specific Sp7 promoters are enriched in genes associated with actin filament assembly (Fig. 4c, right). Systematic comparison of the chromatin landscape[31] at Sp7-bound Ocy-specific enhancer sites revealed that histone modifications surrounding Sp7-bound Ocy-specific enhancers were in largely the same state in both osteocytes and POBs (Fig. 4d). Therefore, Sp7 does not globally change the epigenetic state during the osteoblast-to-osteocyte transition, but rather acts upon open regulatory regions shared between those two cell types (Fig. 4e).

In osteoblasts, Sp7 binds enhancer chromatin indirectly via associating with Dlx family transcription factors;[30] therefore, we asked if Sp7 might utilize a distinct binding cofactor in osteocytes. Comparison of enriched sequence motifs present in each group of enhancers[32] demonstrated that Ocy-specific, shared, and POB-specific enhancers all showed enrichment of binding motifs associated with skeletal development (Supplementary Data 4 show motif enrichment for groups of Sp7-bound peaks, Supplementary Data 5 show gene association for groups of Sp7-bound peaks). As expected[30], POB-specific enhancers are enriched in Dlx-binding sequences (Fig. 4f). In contrast, Ocy-specific enhancers demonstrated selective enrichment for the TGA(G/T)TCA motif bound by AP1 family members (Fig. 4f)[33,34]. Therefore, Sp7 likely cooperates with distinct DNA-binding transcription factors to regulate enhancer activity in osteoblasts versus osteocytes. Next, we intersected RNA-seq and ChIP-seq data sets to identify 77 direct osteocyte-specific Sp7 target genes (Fig. 4g; Supplementary Data 6) for subsequent functional studies.

**Osteocrin rescues morphologic and molecular defects caused by Sp7 deficiency.** Of these 77 direct osteocyte-specific Sp7 target genes, we initially identified top candidates *Ostn*, *Panx3*, and *C1qtnf3* based on the degree of Sp7 regulation and potential to regulate cell morphology changes. Of these genes, CRISPR/Cas9-mediated deletion of the secreted peptide Osteocrin (encoded by the *Ostn* gene)[35] selectively reduced phalloidin staining in 3D culture (Supplementary Fig. 5a–b, here the editing efficiency of each sgRNA, as measured by TIDE[36], is shown next to the corresponding fluorescence-activated cell sorting (FACS) histogram tracing).

The *Ostn* gene encodes a small secreted protein that is expressed in periosteal cells in mouse and human bone, muscle, and primate neurons[35,37]. Supplementary Fig. 6a shows periosteal Ostn expression in control but the reduced expression in Sp7 mutant bones. Osteocrin potentiates C-type natriuretic peptide (CNP) signaling[38,39] in cells at the levels of cGMP production (Ocy454 cells) and downstream ERK1/2 phosphorylation (MC3T3-E1 cells and Ocy454 cells) (Supplementary Fig. 7a, b).

Since *Ostn* is an osteocyte-specific Sp7 target gene whose levels are reduced in *Sp7* knockdown cells (Fig. 3f), we asked if *Ostn* overexpression could rescue defects associated with Sp7 deficiency in vitro. *Ostn* overexpression in *Sp7* knockdown cells led to the rescue of dendrite numbers and F-actin content in 3D culture (Fig. 5a–c). Next, we used RNA-seq (Supplementary Fig. 8a; Supplementary Data 7) where we predicted that genes whose expression was dependent on Sp7 might be 'rescued' by *Ostn* overexpression. Supplementary Fig. 8b shows expected patterns of *Sp7*, *Ostn*, and *Panx3* (a Sp7 target gene whose expression is not regulated by *Ostn*) expression in these experimental conditions. Indeed, 516 out of 560 (92.1%) of the genes differentially regulated in both comparisons (shLacZ vs shSp7 and shSp7: control vs osteocrin overexpression) were counter-regulated by Sp7 and Ostn (Supplementary Fig. 8c). Scatterplot analysis and RRHO2 threshold test demonstrated a strong inverse correlation between profiles seen with *Sp7* knockdown and *Ostn* rescue (Supplementary Fig. 8c).

Next, we asked if increasing systemic Ostn levels might rescue osteocyte defects observed in *Sp7* mutant mice. To do this, we treated $Sp7^{OcyKO}$ mice with an *Ostn*-expressing adeno-associated virus (AAV8, a strategy that increases circulating Ostn levels by boosting hepatic, but not skeletal, Ostn secretion) (Supplementary Fig. 9a, b). AAV8-Ostn treatment rescued defects in dendrite number, dendrite length, and osteocyte apoptosis in $Sp7^{OcyKO}$ mice (Fig. 5d–f, i–k). These AAV8-Ostn effects on osteocyte morphology and survival were associated with reduced intracortical osteoclasts and reduced cortical porosity (Fig. 5g, h, l). In sum, overexpression of the Sp7 target gene *Ostn* rescues the morphologic, transcriptomic, and phenotypic defects seen when Sp7 is absent in mature osteoblasts and osteocytes.

When OSTN occupies the decoy receptor NPR-C, clearance of CNP by NPR-C is reduced, and CNP availability to signal through NPR-B is enhanced. CNP stimulation of NPR-B leads to increases in cGMP production and protein kinase G activation

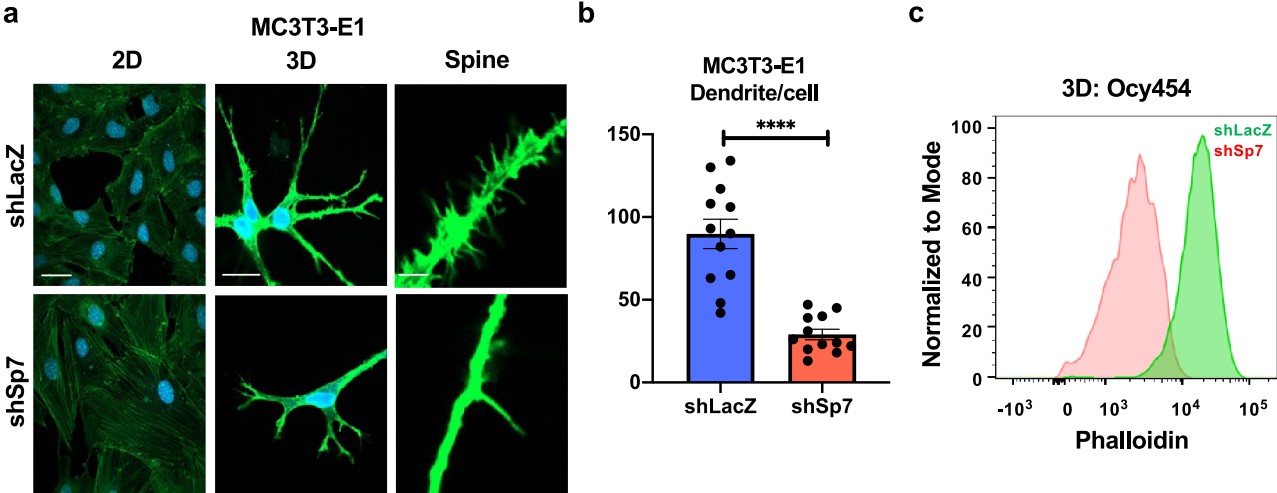

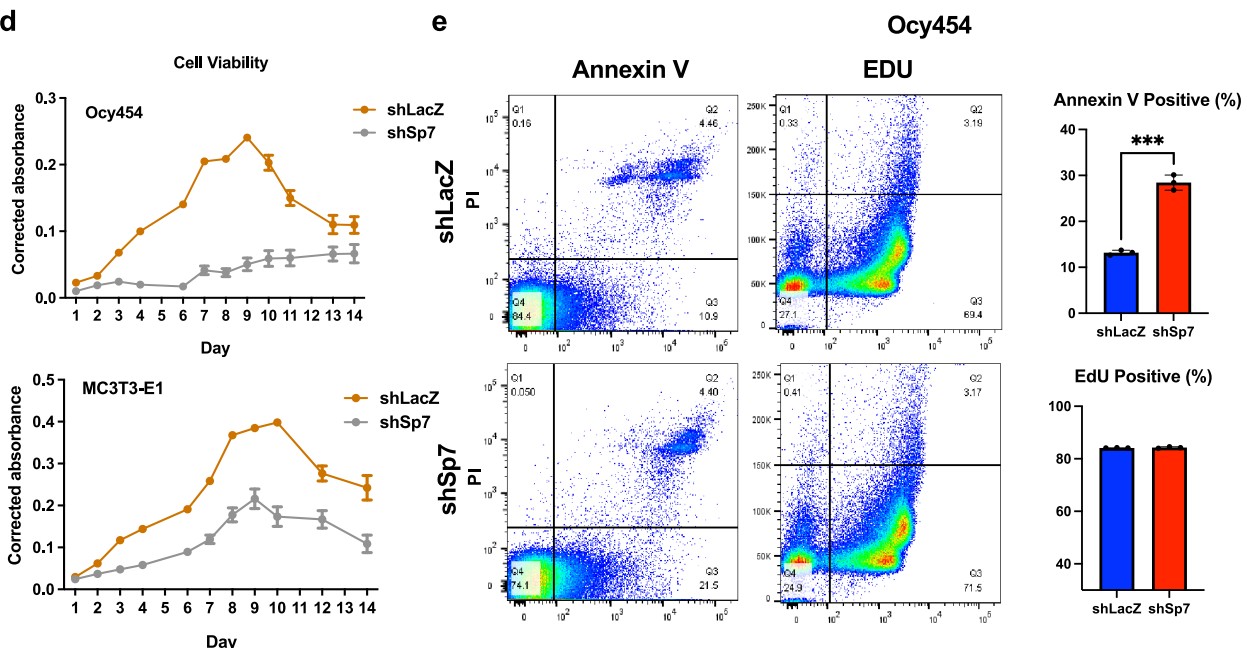

**Fig. 2 Sp7 is required for optimal dendrite formation in vitro. a** Control (shLacZ) and *Sp7* knockdown (shSp7) MC3T3-E1 cells were cultured in standard (2D) or 3D (type I collagen gel) conditions, then stained with phalloidin (green) and DAPI (blue). In 3D culture, *Sp7* knockdown cells show short dendrites (middle panel) with reduced complexity (right panel). Scale bars for 2D and 3D images represent 10 μm, scale bar for "spine" images represent 1 μm. **b** The number of dendrites per cell was measured in *Sp7* knockdown and control cells grown in 3D culture. Each data point represents the dendrite number of an individually measured cell analyzed over three independent experiments. Two-sided student's *t* test was used, **** indicates $p < 0.0001$. **c** Control and *Sp7* knockdown Ocy454 cells were grown in 3D culture then stained with FITC-Phalloidin for flow cytometry. Reduced intracellular phalloidin staining is noted in *Sp7* knockdown cells. **d** Growth curves for control and *Sp7* knockdown Ocy454 cells (top) and MC3T3-E1 cells (bottom) were determined using a resavurin-based viability dye. $n = 3$ biologically independent samples were measured. **e** Control and *Sp7* knockdown Ocy454 cells were analyzed in vitro for apoptosis and proliferation. Normal proliferation is noted based on EdU incorporation, whereas *Sp7* knockdown cells show increased apoptosis. $n = 3$ biologically independent samples were measured. All data are presented as mean values ±SEM. Two-sided student's *t* test was used, *** indicates $p = 0.0001$.

that are enhanced by OSTN[40]. Thus, it is hypothesized that OSTN and CNP synergistically enhance downstream signaling output in target cells. To gain further mechanistic insight into pathways used by OSTN to promote osteocyte dendrite formation, we used Cell Painting[41] to assess the morphologic effects of CNP plus OSTN treatment in osteoblasts (MC3T3-E1 cells) and osteocytes (Ocy454 cells). Control and Sp7 knockdown cells treated for 48 h with the vehicle, low-dose CNP (100 nM), or OSTN (500 nM) were compared with cells treated with both CNP (100 nM) and OSTN. Quantification of 1037 morphology features revealed overall similarities between cells treated with vehicle, CNP alone, or OSTN alone. However, cells treated with CNP plus OSTN had distinct morphological differences from all other groups (Fig. 5m). Moreover, cells treated with Ostn plus low-dose CNP (100 nM) showed overall morphological similarities to cells treated with high-dose CNP

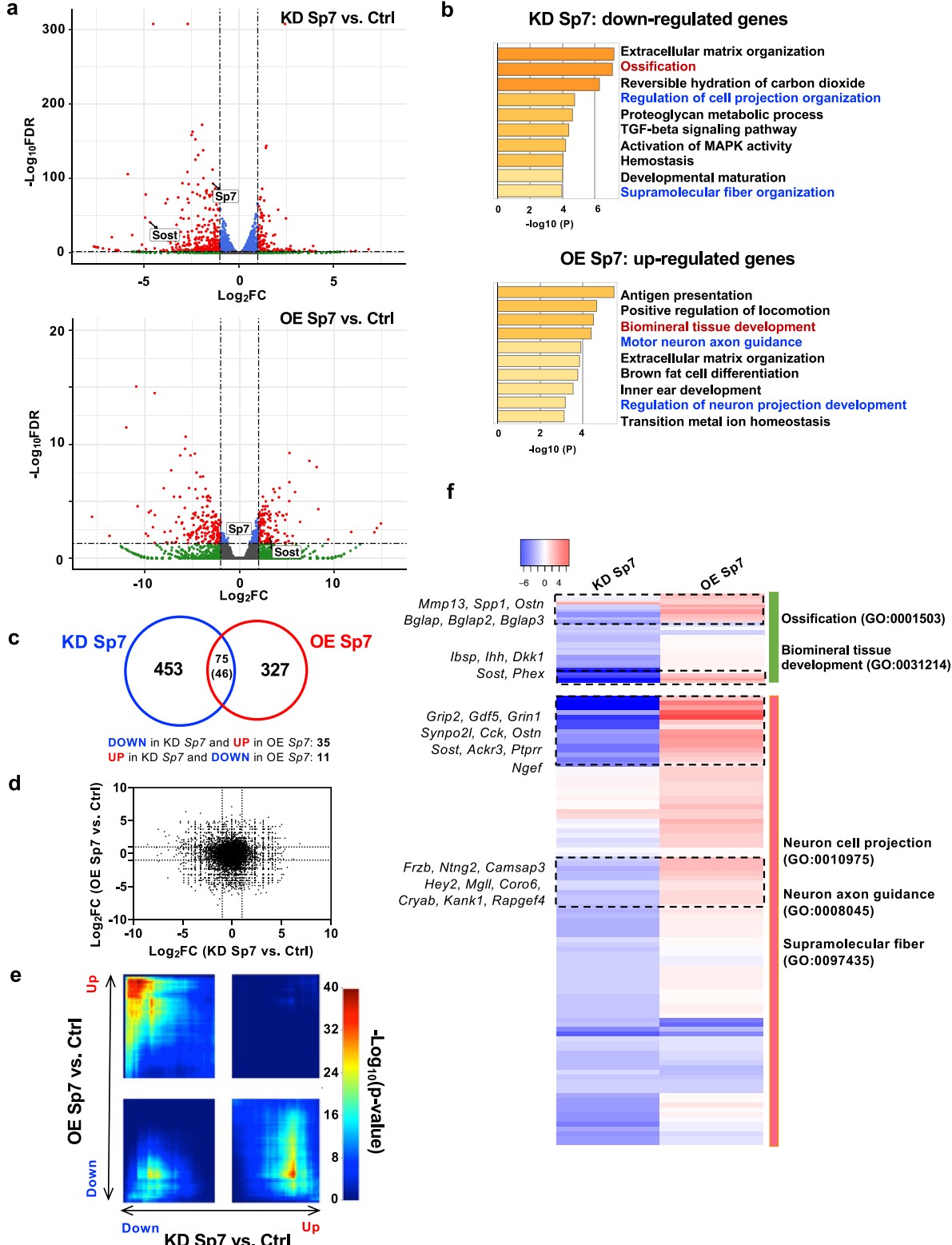

(1000 nM), supporting a mechanism of action of osteocrin via enhancement of CNP signaling. This relationship was apparent in both control MC3T3 and Ocy454 cells (WT MC3T3 con vs OSTN Pearson correlation coefficient = 0.31; WT MC3T3 CNP vs CNP + OSTN Pearson correlation coefficient = −0.31) and shSp7 cells. Further analysis of distinct morphologic features driving differences between cells treated with OSTN + CNP and cells treated with CNP alone support a role for OSTN in

modulating cytoskeletal dynamics (features identified through analysis of phalloidin staining) to change cell shape (Supplemental Fig. 10a–c, Supplemental Data 8).

**Sp7-dependent transitional cells during osteocyte development revealed by single-cell transcriptomics.** The discrete transitional cell types between bone-forming osteoblasts and matrix-embedded,

**Fig. 3 Sp7 controls the expression of genes involved in neuron projection development in osteocytes. a** RNA-seq was performed on control (shLacZ) and *Sp7* knockdown (KD, top) and control (LV-GFP) and *Sp7* overexpressing (OE, bottom) Ocy454 cells. Volcano plots demonstrate differentially expressed genes (red data points) in each comparison. See Supplementary Data 2. **b** Gene ontology analysis of genes downregulated by *Sp7* knockdown (top) and upregulated by *Sp7* overexpression (bottom) reveals enrichment in several terms associated with cell projection development and neuronal morphogenesis. Fisher's exact test with correction for multiple comparisons was used to determine the significance of enriched gene ontology terms. **c–e** Relationship between gene expression changes in response to Sp7 perturbation. Top, Venn diagram revealing number of genes regulated by both *Sp7* overexpression and *Sp7* knockdown. Middle, scatterplot showing fold change regulation of all detected genes by *Sp7* knockdown (x-axis) and *Sp7* overexpression (*y* axis). Bottom, RRHO2 visualization reveals statistically significant groups of genes counter-regulated by *Sp7* knockdown versus overexpression. **f** Heatmap showing fold change regulation of individual genes in gene ontology groups of interest (neuron cell projection and axon guidance).

dendrite-bearing osteocytes are currently unknown. We used *Dmp1-Cre*; tdTomato reporter mice and adapted an aggressive digestion protocol to liberate these cells from the bone matrix. Viable tdTomato+ bone cells from control and *Sp7^OcyKO* were recovered (see Methods) for high-throughput single-cell RNA-seq (scRNA-seq) (Fig. 6). Supplementary Fig. 11 shows gating strategies and quality control metrics. Using LIGER[42], we captured eight distinct *Bglap3*-expressing "Ob/Ocy" osteo-lineage clusters, ranging from proliferating pre-osteoblasts to mature *Sost*-expressing osteocytes (Fig. 6a; Supplementary Fig. 12a; Supplementary Data 9)[42]. Based upon the expression of enriched genes in each cluster, we annotated these 8 groups of cells as (1) *Pdzd2/Bglap3* (canonical osteoblasts), (2) *Pdgfrl/Mgst3*, (3) *Tnc/Enpp1*, (4) *Plxdc2/Postn*, (5) *Tagln2/Enpp1*, (6) *Fbln7/Sost* (mature osteocytes), (7) *Adipoq/Cxcl12* (*Cxcl12*-abundant reticular (CAR) cells; notably others have also reported that Dmp1-Cre labels CAR cells[43]), and (8) *Mki67* (proliferating pre-osteoblasts) (Supplementary Fig. 13). *Tnc, Kcnk2,* and *Dpysl3* mark 3 intermediate subpopulations (clusters 3, 4, and 5, respectively, between osteoblasts and osteocytes (Fig. 6b–d). *Tnc* (Fig. 6b, f) and *Dpysl3* (Fig. 6d, h) are highly expressed in subpopulations of cells in the endosteum and periosteum, and in some early mineralizing osteocytes near bone. *Kcnk2* expression is highest among rare endosteal cells with a canopy-like appearance (Fig. 6c, g). *Fbln7* is expressed in newly formed osteocytes and mature osteocytes buried deep within the bone matrix (Fig. 6e, i).

Comparison of tdTomato+ cells between *Sp7^OcyKO* mice and littermate controls revealed striking changes in cluster cellularity in *Sp7^OcyKO* mice (Fig. 6k–m). *Sp7* mutant mice showed increased relative proportions of cells in transitional clusters 3, 4, and 5 and reduced numbers of mature osteocytes, further demonstrating arrested osteocyte maturation in the absence of Sp7. Next, we performed pseudotime differentiation analysis using Monocle3[44] and Velocyto[45] (Fig. 6n; Supplementary Fig. 12b). These analyses suggest the potential of two differentiation pathways from canonical osteoblasts (cluster 1) via intermediate cells in clusters 3 > 4 > 5 or 2 > 5 into mature *Fbln7/Sost*-expressing osteocytes (cluster 6). Compared with the orderly trajectory seen in WT cells, *Sp7* mutants show multiple defects (red arrows in Fig. 6n, right panel) including apparent arrest in clusters 3 (marked by *Plec*, Supplementary Fig. 14) and 5 (marked by *Lmo2*, Supplementary Fig. 14). Taken together, these single-cell RNA-sequencing results confirm the key role of Sp7 in orchestrating multiple steps in osteocyte differentiation in vivo.

Cluster-specific differential gene expression analysis in intermediate clusters (Supplementary Fig. 12c) revealed dysregulated mineralization-related genes in *Sp7^OcyKO* cells in clusters 3, 4, and 5, including *Enpp1*, *Ank,* and *Mepe* (Supplementary Fig. 13–14). Among dysregulated genes in *Sp7^OcyKO* cluster 5 cells, *Lmo2* marks the expression of a distinct subgroup of cells arrested in the transition to mature osteocytes (Supplementary Fig. 12c and Supplementary Fig. 14). *Sp7^OcyKO* cells show reduced *Pdpn* and *Fbln7* expression in cluster 6, suggesting blocked terminal osteocyte maturation (Supplementary

Fig. 13–14), particularly in light of a role for *Pdpn* in dendrite elongation in osteocytes[46]. Furthermore, Sp7 deficiency is associated with marked dysregulation of terminal osteocyte markers (*Irx5, Dmp1, Ptprz1*) across the entire population of tdTomato+ cells captured. *Ptprz1* is upregulated during osteoblast-to-osteocyte differentiation[28,47]. Although these genes are normally restricted to mature osteocytes, they are aberrantly expressed in multiple subpopulations in *Sp7^OcyKO* mice (Supplementary Fig. 14).

Gene Ontology analysis of the top cluster 6 (osteocytes, marked by enriched expression of *Fbln7/Sost*) marker genes revealed enrichment with "neuronal" terms such as cell projection organization and neuron differentiation (Supplementary Fig. 15a–b). To further explore potential similarity between osteocytes and neurons at the transcriptomic level, we performed enrichment analysis of osteocyte marker genes across cell types in mouse brain[48]. Osteocyte, but not osteoblast, marker genes are significantly enriched in their relative mean expression values in neurons versus other cell types in mouse brain (Fig. 6o). This observation of concordant gene expression patterns between osteocytes and neurons prompted us to perform a similar analysis on a core group of 77 Sp7 target genes as defined from our in vitro RNA-seq and ChIP-seq studies (Fig. 4g). GO analysis of 77 Sp7 targets also showed that they are enriched in terms such as "neuron projection" (Supplementary Fig. 16a). The relative mean expression values of these 77 genes, and a distinct group of 134 POB-specific Sp7 target genes, are significantly enriched in neurons versus other cell types in mouse brain[48] (Fig. 6p; Supplementary Fig. 16b). See Methods and Supplemental Fig. 16 for further details regarding these bone/brain expression analyses. Taken together, these observations demonstrate strong convergence between bone cell Sp7 target genes and neuronally expressed genes.

To further evaluate the similarity between osteocytes and neurons at the level of dendrite formation, we examined the expression "neuronal" cell projection genes (based upon gene ontology annotation) identified from our in vitro analysis in this skeletal single-cell RNA-seq data set. Two categories of "neuronal" Sp7 target genes emerge from this analysis (Supplementary Fig. 18a). One group of genes (exemplified by *Kank1* and *Cryab*) is expressed in osteocytes (c6) and osteoblasts (c1 and 2) in control mice, and downregulated in these clusters in the absence of Sp7 with concomitant induction in 'intermediate' cells in clusters 3 and 5. The second group of genes (exemplified by *Ackr3*, *Ptprr*, and *Sost*—of note, whereas *Sost* expression is largely restricted to osteocytes in bone, this gene has been detected in developing neurons in mice and zebrafish[49,50], thus explaining its inclusion in many neuronal GO groupings) is restricted to osteocytes (c6) in control mice, and downregulated in cluster 6 with subtle upregulation in "intermediate" cluster 5 in Sp7 mutant animals (Supplementary Fig. 18b). This suggests that these neuronal genes may be repurposed to regulate osteocyte dendrite formation downstream of Sp7.

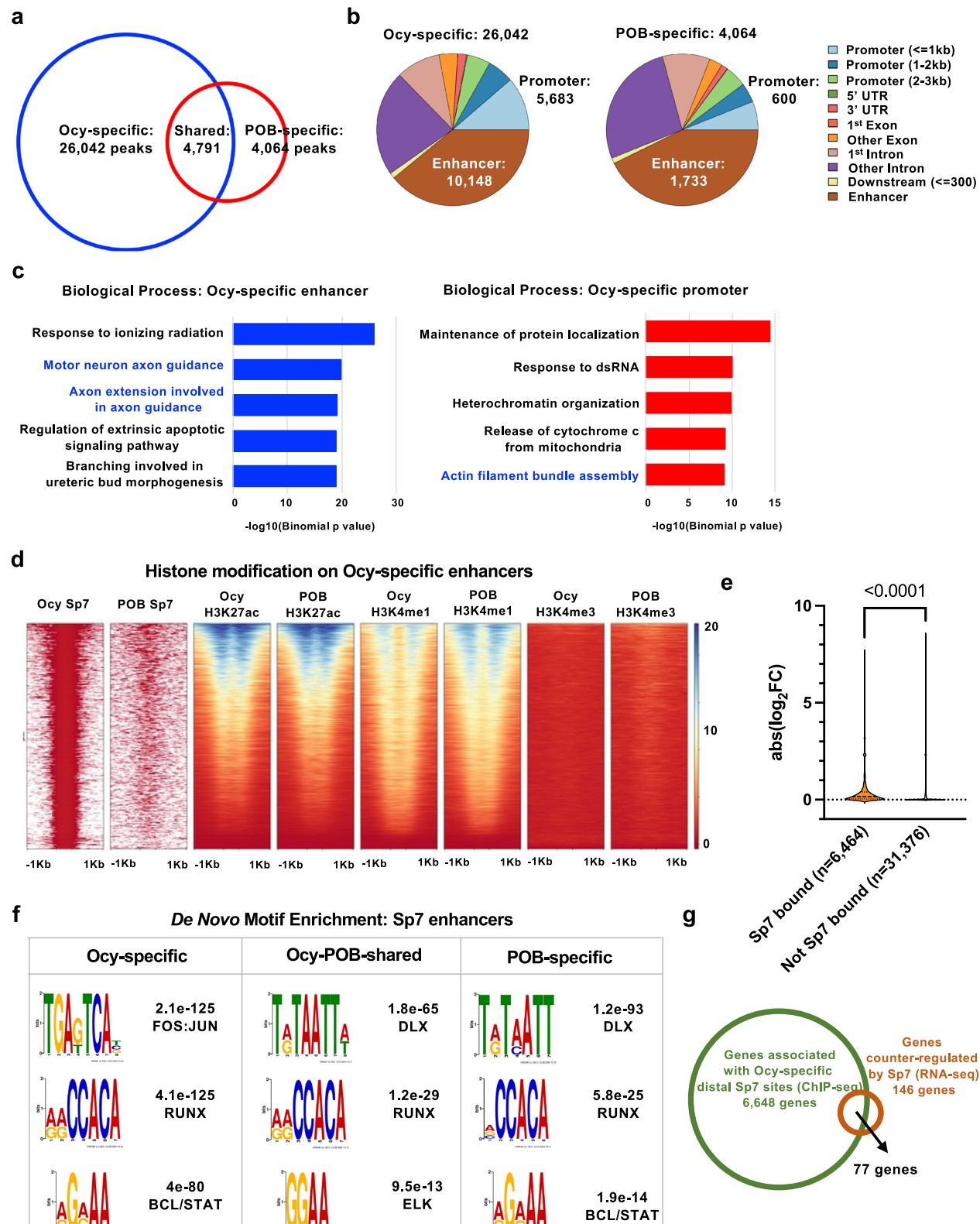

Sp7 expression has been reported during neurogenesis in the olfactory bulb, cortex, and cerebellum. To date, functional studies have only focused on the olfactory bulb, where no obvious defects are noted in germline Sp7-null mice. Consistent with these previous reports[51,52], brain tissue from 8-week-old Sp7-Cre; tdTomato[LSL] mice showed subpopulations of neurons clearly marked with tdTomato expression (Supplemental Fig. 19).

**Sp7 and osteocyte-linked genes are associated with common and rare human skeletal dysplasias.** Next, we sought to further establish the relevance of the osteocyte functions of SP7 in human bone diseases. First, we used MAGMA[53] to understand the relationship between genes enriched in cells undergoing the osteoblast-to-osteocyte transition with genes linked to human BMD variation and fracture risk[13]. This analysis demonstrated

**Fig. 4 Definition of the osteocyte-specific Sp7 cistrome. a** Sp7 ChIP-seq was performed in Ocy454 cells, and Sp7-binding patterns were compared between Ocy454 cells and primary osteoblasts (POB). See also Supplementary Data 3. **b** Genomic distribution of cell type-specific Sp7-binding sites. **c** Gene ontology analysis of genes linked to Ocy454 cell-specific Sp7 peaks. Sp7 associates with regulatory regions of genes linked to motor neuron axon guidance and actin filament bundle assembly. **d** Ocy454 cell-specific Sp7-bound enhancer peaks were analyzed for the indicated histone modifications in d3 (POB) and d35 (Ocy) IDG-SW3 cells. See text for details. **e** Genes were categorized based on the presence of Ocy-specific Sp7 enhancer association in Ocy454 cells. Then, the effect of Sp7 shRNA (versus LacZ shRNA) was reported as the absolute value of the log2 fold change. Genes bound by Sp7 show, on average, a greater effect of Sp7 knockdown than genes without Sp7 enhancer binding sites. Two-tailed Welch's $t$ test was used, $p < 0.0001$. **f** De novo motif analysis of osteocyte-specific, osteoblast-specific, and shared distal Sp7 peaks. See also Supplementary Data 4 and 5. **g** 77 genes (Supplementary Data 6) are revealed by intersecting transcripts regulated by both Sp7 knockdown and overexpression in Ocy454 cells (FDR < 0.05, 146 genes) and genes with associated Ocy454 cell-specific Sp7 ChIP-seq peaks (6,648 genes).

significant enrichment of marker genes identified from osteoblasts and osteocytes, but not other tdTomato+ cells isolated and sequenced in our scRNA-seq analyses, with genes linked to human fracture risk (Fig. 7a). We then examined the enrichment of osteocyte markers in a recent classification of genes that when mutated cause distinct classes of human skeletal diseases[54]. The top osteocyte markers are significantly enriched in other groups of genes that cause "sclerosing bone disorders" and "abnormal mineralization" (Fig. 7b).

Patients homozygous for $SP7^{R316C}$ present with osteogenesis imperfecta-like bone disease including multiple fragility fractures and short stature[14,15]. Sp7 binds an enhancer ~110 kB upstream of the *Ostn* transcription start site (Fig. 7c). The transcriptional activity of this putative *Ostn* enhancer responded to *Sp7* overexpression in heterologous reporter assays (Supplemental Fig. 20a). Although both SP7 cDNAs were equally expressed (Fig. 7d), $SP7^{R316C}$ was defective at transactivating this *Ostn* reporter element (Fig. 7e).

Similar to $Sp7^{OcyKO}$ mice (Fig. 1b), bone biopsies from $SP7^{R316C}$ patients show high cortical porosity and increased bone turnover[15]. We analyzed osteocyte morphology in non-decalcified iliac crest bone biopsies from two $SP7^{R316C}$ patients and two age and sex-matched healthy patients. Using a silver staining protocol (see Methods), we observed reduced dendrite length and number in $SP7^{R316C}$ patients versus controls (Fig. 7f–h). As $SP7^{R316C}$ patients can stand upright and survive into adolescence (unlike complete *Sp7*-null mice which die just after birth owing to lack of osteoblasts[11]), our results suggest that this hypomorphic *SP7* mutation selectively interferes with the function of Sp7 in osteocytes. Consistent with this model, whereas overexpression of both WT and $SP7^{R316C}$ constructs similarly increase expression of osteoblast marker genes (*Bglap* and *Alpl*) in Ocy454 cells, only WT SP7 stimulates expression of mature osteocyte marker genes (*Sost*) and genes related to dendrite formation (*Ostn*) (Fig. 7i).

## Discussion

Here, we report a key role for the transcription factor Sp7 in osteocytogenesis, the process through which some osteoblasts embed themselves within the mineralized bone matrix and become osteocytes. Sp7 has a key role in early skeletal progenitors and their commitment to the osteoblast lineage[11]. Sp7 controls osteoblast-specific gene expression via directly binding to GC-rich promoter sequences[55] and indirect DNA binding in conjunction with Dlx family transcription factors[30] and NO66 histone demethylases[56]. However, *Sp7* expression persists throughout the osteoblast lineage[57], and our current findings demonstrate an important role for this transcription factor in osteocytogenesis. In osteocytes, Sp7 associates with a distinct distal enhancer DNA motif (Fig. 4f), suggesting that unique Sp7 modifications and/or cofactors explain the cell type-specific function of this transcription factor at different stages in the osteoblast lineage.

Although "master regulator" transcription factors have been identified for other cell types in bone[10], factors responsible for osteocyte-specific gene expression patterns have proved elusive. Mef2 family transcription factors, which also play key roles in growth plate chondrocytes[58] and neurons[59], drive expression of the osteocyte-restricted transcript *Sost*[60–62]. However, mice lacking Mef2c in osteocytes show high bone mass but normal osteocyte morphology. Loss of Hox11 function in adult mice at the osteoblast-to-osteocyte transition causes defects in osteocyte morphology similar to those seen in our $Sp7^{OcyKO}$ mice[63]. The molecular mechanism through which Hox11 factors control osteocyte morphology remains unknown. Here, we demonstrate that Sp7 has a vital role in orchestrating osteocyte differentiation and identifying cell type-specific target genes responsible for this effect. Although a previous report suggested a role of Sp7 in osteocyte morphology[25], our study employs transcriptomic, epigenomic, and single-cell profiling in order to define the molecular mechanism used by Sp7 to control osteocyte maturation, and shows the clinical relevance of this role.

Relatively few osteocyte-specific Sp7 target genes were identified by intersecting our transcriptomic and ChIP-seq data (Fig. 4g). Of these genes, *Ostn* overexpression rescues Sp7-deficient phenotypes in vitro and in vivo. Osteocrin was initially identified based on its expression in osteoblasts and early embedding osteocytes[35,37]. However, the biological functions of osteocrin have mainly been studied in the growth plate, muscle, and heart[40,64–67]. Of note, primate, but not rodent, neurons express *Osteocrin* in an activity-dependent manner, and *osteocrin* expression in neurons of higher species may be linked to increased synaptic complexity[39]. Osteocrin is thought to enhance C-type natriuretic peptide receptor signaling by targeting NPR3, a CNP clearance receptor, for degradation. In doing so, osteocrin enhances CNP signaling via NPR2 and increases intracellular cGMP levels. While the downstream signaling mechanisms used by osteocrin to promote osteocytogenesis remain unknown, it is interesting that cGMP has been linked to osteocyte survival downstream of semaphorin 3 A signaling[68]. A recent report demonstrates that periosteal osteocrin expression is regulated by mechanical cues;[69] therefore, investigating whether the Sp7/Ostn axis described here participating in osteocytic adaptation during mechano-transduction represents a priority for future study. Although our current model suggests that Ostn acts downstream of Sp7 in promoting osteocyte maturation, we acknowledge that future studies are needed to examine a potential role of Sp7 in controlling gene expression changes downstream of Ostn signaling in normal osteocyte differentiation.

Defects in osteocyte dendrites in $Sp7^{OcyKO}$ mice are likely to be the cause, not effect, of increased osteocyte apoptosis (Fig. 1r, v). Aging and pharmacologic glucocorticoid exposure are two common conditions in which defects in osteocyte morphology and increased osteocyte apoptosis are noted. Whether osteocrin administration may rescue defects in these pathologic states remains to be determined. Our model of osteocrin administration

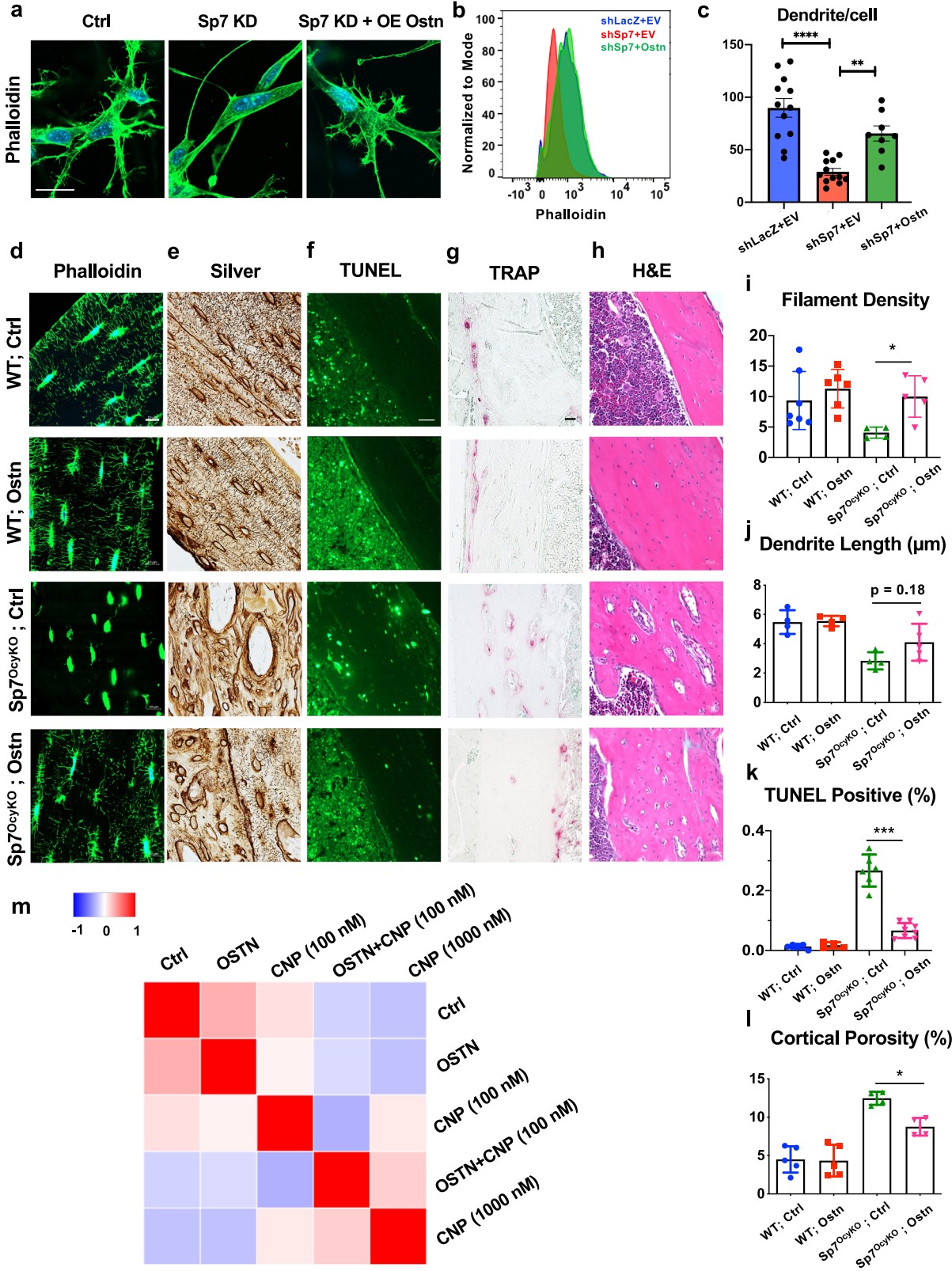

cannot discriminate between the effects of this peptide on dendrite formation versus maintenance. In *Sp7^{OcyKO}* mice, it is likely that reduced intercellular connectivity leads to increased osteocyte apoptosis. Osteocyte apoptosis triggers RANKL-driven osteoclast recruitment;[70,71] therefore, increased intracortical remodeling (Fig. 1c–d) observed in *Sp7^{OcyKO}* mice is likely due to primary defects in osteocyte morphology. Although we favor the

model that defects in osteocyte dendrites are the primary insult that ultimately drives the cortical bone phenotype in *Sp7^{OcyKO}* mice, we acknowledge the possibility that other molecular functions of Sp7 in mature osteoblasts and osteocytes may contribute.

Both MC3T3-E1 (Fig. 2a) and Ocy454 (Supplemental Fig. 3a) cells grown in 3D collagen gels develop complex dendrite projections in an Sp7-dependent manner. Transcriptomic analysis of

**Fig. 5 Exogenous osteocrin rescues skeletal phenotypes associated with Sp7 deficiency. a–b** *Sp7* knockdown (KD) MC3T3-E1 cells were infected with control (empty vector) or osteocrin (LV-Ostn) lentiviruses followed by growth in 3D collagen gels and phalloidin (green) and DAPI (blue) staining. *Osteocrin* overexpression rescues phalloidin staining intensity to levels observed in control (shLacZ) cells. Scale bar represents 10 μm. **c** Dendrites number per cell was measured in the indicated cells grown in 3D collagen gel. Each data point represents the dendrite number of individual cells analyzed over three independent experiments. Data are presented as mean values ±SEM. Fisher's exact test with correction for multiple comparisons was used to determine the significance of enriched gene ontology terms. Ordinary one-way ANOVA was used followed by Tukey's multiple comparisons test. **** indicates $p < 0.0001$. ** indicates $p = 0.0036$. **d–h** Control or $Sp7^{OcyKO}$ mice were injected with AAV8-control or AAV8-osteocrin viral particles at 3 weeks of age, then analyzed 3 weeks later for histologic analysis by phalloidin staining (**d**), silver staining (**e**), TUNEL (**f**), TRAP (**g**), or H & E (**h**). The scale bar in **d–e** is 10 μm. The scale bar in **f–h** is 50 μm. All results are quantified in **i–l**. In these panels, each data point represents the indicated parameter measured in a biologically independent animal. Data are presented as mean values ±SEM. Ordinary one-way ANOVA with Dunnett's correction was used to determine significance. Exact $p$ values are: **i** 0.0292, **j** 0.178, **k** <0.0001, **l** 0.0236. **m** Cell painting similarity matrix of MC3T3-E1 cells treated with control, human OSTN peptide (500 nM), low-dose CNP (100 nM), OSTN (500 nM)+CNP (100 nM) and high-dose CNP (1000 nM). Cells treated with CNP plus OSTN and with high-dose CNP had distinct morphological differences from all other groups.

3D versus 2D cultures (Supplemental Fig. 4a) demonstrates that 3D culture recapitulates many aspects of osteocyte maturation. Careful inspection of dendritic projections in both cell types reveals the presence of fine 'spine'-like protrusion emanating perpendicular to the main axis of these extensions (Fig. 2a). At present, we do not understand the significance of these spine-like projections, though we speculate that these structures may be similar to "tethering elements" described at the ultrastructural level used by bona fide osteocytes to connect to bone surfaces and amplify mechanical cues[72].

Previous efforts to characterize non-hematopoietic cells in bone using single-cell profiling have failed to capture significant numbers of cells at the osteoblast-to-osteocyte transition[73–76]. Our approach combining Dmp1-Cre; tdTomato reporter mice with serial enzymatic digestions allowed us to identify a population of cells for single-cell profiling enriched in mature osteoblasts, differentiating osteocytes, and some mature osteocytes. This population of cells is enriched in the expression of genes previously linked to human fracture risk (Fig. 7a), highlighting the key contribution of matrix-associated/embedded cells to human bone mass/strength. Multiple osteoblast/osteocyte subset markers were identified in this work (Fig. 6; Supplementary Fig. 13). Sp7 deficiency leads to an accumulation of mature osteocyte precursors and a relative paucity of mature osteocytes, an observation confirmed by complementary analyses designed to interrogate the cellular differentiation stage. Future studies are needed to use the markers identified here to define the location and morphologic features of distinct osteocyte precursor subsets in situ.

Dendritic projections allow osteocytes to quickly communicate with each other via gap junctions in order to sense external cues that in turn regulate bone homeostasis[77]. Although devoid of action potentials, osteocytes use calcium fluxes for electric coupling between cells to facilitate rapid response to mechanical cues[78]. As such, potential parallels between highly connected osteocytes and neurons may emerge. Osteocyte-specific Sp7 target genes and osteocyte-enriched transcripts are enriched in neurons versus other cell types in the brain (Fig. 6o, p), likely due to common requirements for cytoskeletal factors in dendritic projections of osteocytes and neurons. While a great deal is known about the constituents of neuronal projections including axons, dendrites, and dendritic spines and the external cues that regulate neuronal connectivity[79], the transcriptional programs that drive axon and dendrite formation during neuronal development and adult neurogenesis remain less well understood[80]. Whether the Sp7-dependent gene module identified herein osteocytes also participate in neuronal morphogenesis remains to be determined. Shared patterns of gene expression between osteocytes and neurons are likely to reflect the common use of fundamental cytoskeletal factors to maintain cellular projections. It is possible that the Sp7-dependent signature identified here also participates in

the development and maintenance of cellular projections in "dendrite"-bearing cell types in other organs[81].

Humans bearing homozygous $SP7^{R316C}$ mutations display skeletal fragility with a phenotype reminiscent of $Sp7^{cKO}$ mice. Notably, this point mutation appears to selectively interfere with the osteocytic functions of Sp7. Future studies are needed to assess the effects of this point mutation on Sp7 DNA binding at the genome-wide level and to understand how this mutation interferes with the osteocyte-specific functions of this transcription factor.

Taken together, our findings demonstrate that Sp7, a gene linked to rare and common human skeletal traits, orchestrates osteocytogenesis via a suite of target genes that promote the optimal formation and maintenance of osteocyte dendrites. Osteocrin, a secreted factor positively regulated by Sp7, can rescue osteocyte morphology and survival defects in Sp7-deficient mice. These findings highlight shared features between osteocytic and neuronal connectivity and highlight steps in osteocytogenesis that may be targeted to improve bone strength for individuals with osteoporosis.

## Methods

**Mice.** *Dmp1-Cre* (9.6 kB) transgenic mice[22] (RRID: MGI:3784520) and Ai14 Cre-dependent tdTomato reporter (JAX, catalog number 007914) were intercrossed. Floxed *Sp7* mice were kindly provided by Dr. Benoit de Crombrugghe[82]. *Dmp1-Cre; tdTm+* mice were crossed with *Sp7fl/+* mice to generate $Sp7^{OcyKO}$ (*Dmp1-Cre; Sp7fl/fl; tdTm+*) and control (*Dmp1-Cre; Sp7+/+; tdTm+*) mice. Genotypes were determined by PCR using primers listed in Supplementary Data 10. All mouse strains were backcrossed to C57BL/6 J for at least four generations. Although this degree of backcrossing may be insufficient for subtle skeletal phenotypes, the phenotype of $Sp7^{OcyKO}$ is quite striking. Moreover, littermate controls were used for all studies. Both males and females were included in this study. All procedures involving animals were performed in accordance with guidelines issued by the Institutional Animal Care and Use Committees (IACUC) in the Center for Comparative Medicine at the Massachusetts General Hospital and Harvard Medical School under approved Animal Use Protocols (2019N000201). All animals were housed in the Center for Comparative Medicine at the Massachusetts General Hospital (21.9 ± 0.8 °C, 45 ± 15% humidity, and 12-h light cycle 7 am–7 pm).

**Cell culture.** Cells were passed in alpha-minimum essential medium (MEM) supplemented with heat-inactivated 10% fetal bovine serum and 1% antibiotic–antimycotic (Gibco™) with 5% $CO_2$. Ocy454 cells[21] were plated at $10^5$ cells ml$^{-1}$ to reach confluence at the permissive temperature (33 °C) in 2–3 days. Ocy454 cells express a thermosensitive large T antigen which is active at 33 °C and inactive at 37 °C. Subsequently, cells were switched to the non-permissive temperature (37 °C) to promote osteocyte differentiation. For protein and gene expression analyses, cells were analyzed after culture at 37 °C for 14–21 days. For microscopy and flow cytometry analyses, cells were cultured at 37 °C for 5–7 days prior to staining. MC3T3-E1 cells (subclone 4, ATCC CRL-2593) were plated at $10^5$ cells ml$^{-1}$ to reach confluence and were kept at 37 °C for the indicated time points.

For 3D culture, rat-tail type I collagen (Advanced BioMatrix, 5153) was first mixed with the neutralized buffer on ice (neutralized buffer: 1.7 ml of 3 M NaOH, 10 ml of 1 M HEPES, 1.1 g of NaHCO$_3$, up to 50 ml of dH$_2$O) to reach pH = 7. Cells were diluted to $5 \times 10^5$ cells ml$^{-1}$ before mixing with the collagen-buffer mix on ice (1:1 volume). The cell/collagen/buffer mixture was plated in 12-well or 24-well plates and incubated at 37 °C for 10 min to polymerize. An equal volume of culture media (alpha-MEM supplemented with heat-inactivated 10% fetal bovine

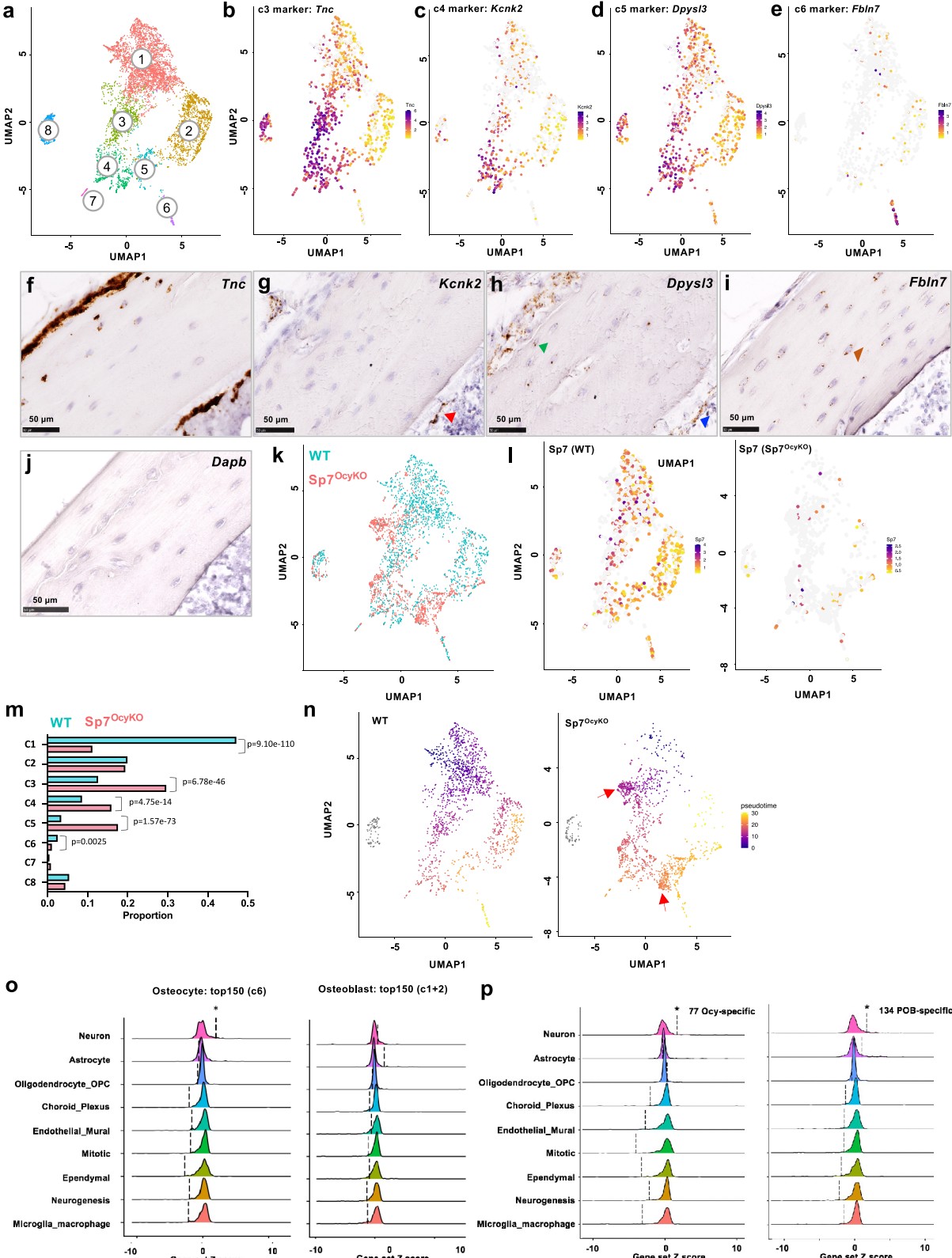

serum and 1% antibiotic–antimycotic, Gibco™) was added on top of the solidified gel prior to subsequent culture.

**shRNA infection and lentiviral transduction**. See Supplementary Data 8 for all shRNA sequences used. For shRNA, lentiviruses were produced in HEK293T cells in a pLKO.1-puro (Addgene, plasmid 8453) backbone. Viral packaging was performed in 293T cells using standard protocols (http://www.broadinstitute.org/rnai/public/resources/protocols). In brief, HEK293T cells were plated at $2.2 \times 10^5$ ml$^{-1}$ and transfected the following day with an shRNA-expressing plasmid (shSp7 or shLacZ plasmid) along with psPAX2 (Addgene, plasmid 12260) and pMD2.G (Addgene, plasmid 12259) using Fugene-HD (Promega). The medium was changed the next day and collected 48 h later. Ocy454 cells were exposed to lentiviral particles overnight at 33 °C in the presence of polybrene (2.5 µg ml$^{-1}$).

**Fig. 6 Single-cell transcriptomic profiling of mature osteoblasts and differentiating osteocytes highlights a key role for Sp7. a** Long bones were subjected to serial collagenase/EDTA digestions (see methods), and cells from fractions 5, 7, and 8 were collected for flow cytometry. Viable (DAPI-negative) tdTomato-positive cells were sorted followed by single-cell RNA-seq library construction. See also Supplementary Figs. 5 and 6. Cell-clustering from WT mice was performed. **b–e** Feature plots showing the expression of cluster-specific markers: *Tnc* (c3); *Kcnk2* (c4); *Dpysl3* (c5); *Fbln7* (c6). **f–j** RNA in situ hybridization of cluster-specific markers: *Tnc, Kcnk2, Dpysl3,* and *Fbln7*. *Kcnk2* expression (red triangle) is primarily noted in endosteal cells that form a canopy around the osteoblast. *Dpysl3* expression is noted in osteocytes close to bone surfaces (green) and a subpopulation of endosteal cells (blue). *Fbln7* expression is noted only in embedded osteocytes (brown). Negative control: *Dabp*. **k** Integrated analysis of cells from WT (blue) and *Sp7^OcyKO^* (red) mice after down-sampling of WT library to match cellular representation seen in *Sp7* mutant mice. **l** Feature plot showing *Sp7* expression across 8 Ob/Ocy clusters. As expected, *Sp7* expression is reduced in mutant mice. **m** Relative proportions (out of 1.0) in WT vs *Sp7^OcyKO^* mice. *Sp7* mutants show reduced canonical osteoblasts (cluster 1), increased cells in clusters 3–5, and reduced cells in cluster 6 (mature osteocytes). Barnard's exact test was used to determine the significance between genotypes. **n** Monocle3 analysis showing pseudotime trajectory mapping across tdTomato-positive cells in WT (left) and *Sp7^OcyKO^* (right) mice. Red arrows indicate Sp7-specific subpopulations in clusters 3 and 5, demonstrating apparently arrested differentiation. **o** Left: the expression of top 150 mature osteocyte markers (c6) was analyzed in a mouse brain single-cell RNA-seq atlas, where significant enrichment is found in aggregated expression values for neurons ($p = 0.022$) as compared against all other major cell type aggregated expression values. Right: the expression of the top 150 osteoblast marker (c1 + 2) was analyzed in a mouse brain single-cell RNA-seq atlas. *OPC* oligodendrocyte progenitor cell, *Mitotic* mitotic cells, *Neurogenesis* neurogenesis-associated cells. **p** Left: the expression of 77 osteocyte-specific Sp7 targets was analyzed in a mouse brain scRNA-seq atlas, where significant enrichment is found in aggregated expression values for neurons ($p = 0.047$) as compared against all other major cell type aggregated expression values. Right: the expression of 134 POB-specific Sp7 targets was analyzed in a mouse brain scRNA-seq atlas, where significant enrichment is found in aggregated expression values for neurons ($p = 0.045$) as compared against all other major cell type aggregated expression values. Wilcoxon rank-sum test was used in **o** and **p**, exact *p* values are listed above.

Media was then changed with puromycin ($4 \, \mu g \, ml^{-1}$). MC3T3-E1 cells were exposed to lentivirus overnight at 37 °C. Cells were maintained in a selection medium throughout the duration of the experiment.

Mouse FLAG-Sp7 or osteocrin cDNAs were introduced via lentivirus via pLX_311 backbone (Addgene, plasmid 118018) as previously described[21]. Briefly, mouse Sp7 and green-fluorescent protein (GFP) lentiviruses were generated by transfecting HEK293T cells with a blasticidin resistance backbone (Addgene, plasmid 26655) along with psPAX2 and MD2.G. Ocy454 and MC3T3-E1 cells were infected with lentiviral particles expressing GFP or Sp7. After 24 h, cells were selected with blasticidin ($4 \, \mu g \, ml^{-1}$) and used for subsequent experiments. To overexpress *Ostn* in *Sp7* knockdown cells, *Sp7* knockdown cells were exposed to Ostn or EV lentiviral particles overnight at 37 °C in the presence of polybrene ($2.5 \, \mu g \, ml^{-1}$). Media was changed with puromycin ($4 \, \mu g \, ml^{-1}$) and blasticidin ($4 \, \mu g \, ml^{-1}$) on the next day. FLAG-hSP7, FLAG-hSP7^R316C^, and EF1a-eGFP constructs were synthesized de novo (VectorBuilder).

**AAV8 infections**. AAV8 vectors encoding codon mouse *Ostn* or GFP under the control of the chicken beta-actin (CAG) promoter (AAV8-CAG-mOstn-WPRE and AAV8-CAG-eGFP vectors) were generated (Vector Biolabs). In all, 3-week-old mice were injected with AAV8 by intraperitoneal (IP) injection at a dose of $5 \times 10^{11}$ gc per mouse in a total volume of 100 μl.

**Human bone sample silver impregnation and osteocyte histomorphometry**. Control bone samples were obtained from the bone biopsies bank of the LIM 16-Laboratório de Fisiopatologia Renal, Hospital das Clínicas HCFMUSP, Faculdade de Medicina-Universidade de São Paulo[83]. Control patient 1 (Ctrl1) is an 18-year-old Caucasian male, and control patient 2 (Ctrl2) is a 14-year-old mixed-race (African and Portuguese) male. Patients with *SP7^R316C^* mutations are as described in[15]. The Sao Paulo University ethics committee (664/97) approved studies for control sample collection. Patient 1 (P1) was a 14-year-old male of Iraqi descent, and patient 2 (P2) is a 12-year-old male of Iraqi descent. All affected family members or their legal guardians provided written informed consent and The Sydney Children's Hospital Network Human Research Ethics Committee (CCR2017/19) approved all studies with P1 and P2. Transiliac bone biopsies were performed 3–5 days after a course of double labeling with tetracycline ($20 \, mg \, kg^{-1}$ per day for 3 days) with a 10-day interval. Undecalcified specimens were fixed in 70% ethanol, dehydrated, embedded in methyl methacrylate, and cut using a tungsten carbide knife. In all, 5 μm-thick sections were stained with Goldner's Trichrome for static bone parameters. Silver nitrate impregnation was used for osteocytes lacunae and its dendrites appreciation. The sections were deacylated in cold MMA for 10–15 min. After rehydration in graded alcohol solutions, 5 μm-thick sections were decalcified in a 20% ethylenediaminetetraacetic acid (EDTA; pH = 7.8) for 30 min and incubated with 200 mM silver nitrate in a solution containing 0.6% gelatin type A and 0.3% of formic acid for 30 min. The sections were then washed and developed in an aqueous 15% solution of sodium thiosulfate ($Na_2S_2O_3$) for 10 min. Unstained 10-μm-thick sections were analyzed under UV light for dynamic parameters. Quantification of the dendrite number and length were performed with ImageJ by thresholding grayscale images for dark, silver-stained lacunae and canaliculi. All histomorphometric analyses were performed using a semi-automatic image analyzer and OsteoMeasure software (OsteoMetrics, Inc., Atlanta, GA, USA), at ×125 magnification, and the full bone structure located between the two cortical areas was evaluated.

**Histology and immunohistochemistry**. Formalin-fixed paraffin-embedded decalcified tibia sections from 6-week-old and 8-week-old mice were obtained. For anti-Sp7 immunohistochemistry staining, antigen retrieval was performed using proteinase K ($20 \, \mu g \, ml^{-1}$) for 15 min. Endogenous peroxidases were quenched, and slides were blocked in TNB buffer (PerkinElmer), then stained with anti-Sp7 antibody (Abcam, ab22552) at a concentration of 1:200 for 1 h at room temperature. For activated caspase-3 IHCs, sections were stained with primary antibody (Cell Signaling Technology, 12692) at 1:500 overnight at 4 °C. Sections were washed, incubated with horseradish peroxidase (HRP)-coupled secondary antibodies, signals amplified using tyramide signal amplification, and HRP detection was performed using 3,3′-diaminobenzidine (DAB, Vector Laboratories) for 2–3 min. Slides were briefly counterstained with hematoxylin before mounting. Hematoxylin and eosin staining was performed using standard protocols. Quantification of Sp7-positive osteocytes and empty lacunae were performed in ImageJ imaging software in a blinded manner.

**Immunoblotting and ELISA**. Immunoblotting was performed as previously described[84,85]. MC3T3 and Ocy454 cells were plated at $10^5$ cells/ml to reach confluence, and Ocy454 cells were differentiated for 7 days at 37 °C. Cells were serum-starved for 4 hours prior to drug treatment with hOstn peptide and/or hCNP (Sigma N8768) for 15 min. For both Ocy454 and MC3T3-E1 *Sp7* knockdown and overexpressing cells, whole-cell lysates were prepared using TNT (Tris-NaCl-Tween buffer, 20 mM Tris-HCl pH = 8, 200 mM NaCl, 0.5% Triton X-100 containing protease inhibitor (PI), 1 mM NaF, 1 mM DTT, 1 mM vanadate). Adherent cells were washed with ice-cold phosphate-buffered saline (PBS), then scraped into TNT buffer on ice. The material was then transferred into Eppendorf tubes kept on ice, vortexed at top speed for 30 s, then centrifuged at top speed for 6 min at 4 °C. For immunoblotting, lysates were separated by sodium dodecyl sulphate–polyacrylamide gel electrophoresis, and proteins were transferred to nitrocellulose. Membranes were blocked with 5% milk in tris-buffered saline plus 0.05% Tween-20 (TBST) and incubated with primary antibody overnight at 4 °C diluted in TBST plus 5% bovine serum albumin BSA. The next day, membranes were washed, incubated with appropriate HRP-coupled secondary antibodies (Anti-rabbit HRP, Cell Signaling Technology 7074, 1:2000), and signals were detected with ECL Western Blotting Substrate (Pierce) or ECL Plus Western Blotting Substrate (Pierce). The primary antibodies were Sp7 (Abcam, ab22552, dilution 1:1000), DYKDDDDK tag (Cell Signaling Technology, 2368, dilution 1:1000), GAPDH (Cell Signaling Technology, 2118, dilution 1:1000), ERK1/2 (Cell Signaling Technology, 4695, dilution 1:1000), phospho-ERK1/2 (Cell Signaling Technology, 9101, dilution 1:2000) and β-tubulin (Cell Signaling Technology, 5346, dilution 1:1000). For transfected HEK293T cells, protein lysates were collected 48 h after transfection as above.

**Sclerostin ELISA**. Sclerostin ELISAs were performed using an antibody pair (Scl Ab-VI and Scl Ab-VII) as described previously[21]. Conditioned medium (48–72 h) was harvested from Ocy454 cells as indicated in the figure legends and stored at −80 °C until further use. High binding 96-well plates (Fisher, 21-377-203) were coated with Scl Ab-VI capture antibody ($3 \, \mu g \, ml^{-1}$) in PBS for 1 h at room temperature. Plates were washed (PBS plus 0.5% Tween-20) and blocked with wash buffer supplemented with 1% BSA and 1% normal goat serum for one hour at room temperature. Samples ($60 \, \mu l \, well^{-1}$) were then added along with a standard curve of murine recombinant Sclerostin (Alpco, Salem, NH) and plates were

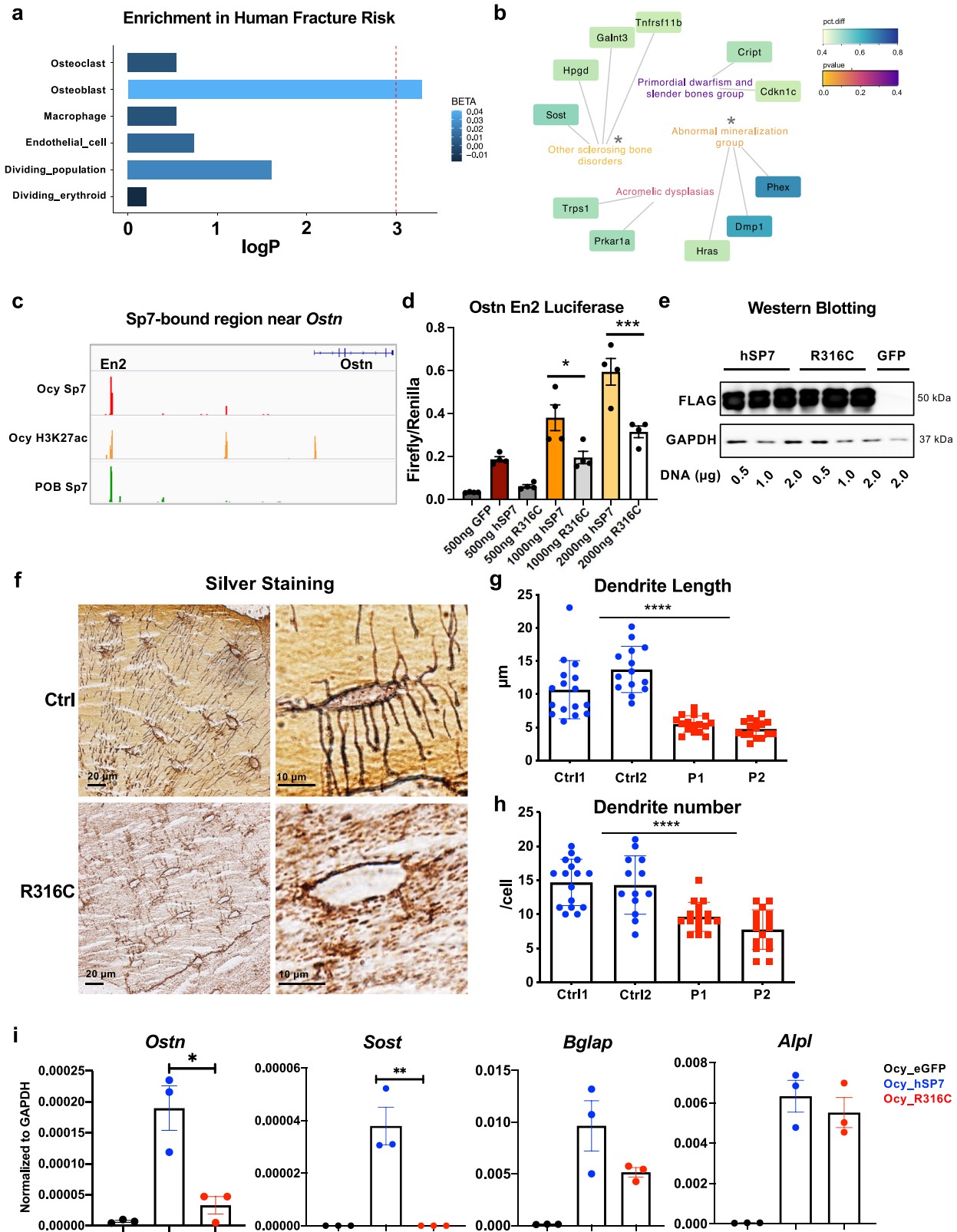

incubated overnight at 4 °C. Plates were washed three times and then incubated with HRP-coupled Scl Ab-VII detection antibody (0.5 µg ml⁻¹) for 1 h hour at room temperature. After washing, signal detection was performed using Ultra TMB ELISA (Pierce, Rockford, IL), stopped by 2 N sulfuric acids, and read at 450 nm. Prior to harvesting supernatant, cell number per well was always determined using PrestoBlue assay (Life Technology) read at 570 nm and 600 nm according to the manufacturer's instructions.

**Measurements of intracellular cGMP.** Ocy454 cells, grown in 96-well plates, were incubated at 37 °C for 2 days. Cells were subsequently stimulated with an incubation medium (Hank's Balanced Salt solution with 0.075% BSA, 10 mM HEPES, and 2 mM IBMX) containing 50 nM human CNP or 500 nM human OSTN at room temperature for 30 min. Cells were collected in 0.1 M HCl. The amount of cGMP in each cell lysate was measured using the Cyclic GMP ELISA Kit (Cayman Chemical) according to the manufacturer's protocol.

**Fig. 7 Sp7 and osteocyte-linked genes are associated with common and rare human skeletal diseases. a** Negative log Bonferroni-corrected $p$ values obtained from the MAGMA gene-set analyses for all major cell types sampled from WT mice, shaded by effect size (BETA) values from the MAGMA gene-set regression. See Methods for a detailed description of the statistical methods used. **b** Association between skeletal dysplasia disease groups and genes enriched in the top mature osteocyte markers. Two disease groups are significantly enriched: "Other sclerosing bone disorder" ($p = 0.016$) and "abnormal mineralization" group ($p = 0.039$). **c** Locations of Sp7 and H3K27ac binding in the osteocrin regulatory region in Ocy454 cells and primary osteoblasts (POB). **d–e** Ostn_En2 activity is induced by WT, but not R316C, FLAG-tagged *SP7* overexpression in HEK293T cells. Comparable expression of both FLAG-tagged SP7 versions is noted by immunoblotting, with GAPDH as a loading control. **d** each data point represents luciferase activity measured in a biologically independent sample. Data are presented as mean values±SEM. Ordinary one-way ANOVA followed by multiple comparisons test was used. *$p = 0.0235$, ***$p = 0.0004$. **f–h** Non-decalcified iliac crest biopsy samples from age/sex-matched and *Sp7*$^{R316C}$ patient samples were silver-stained to assess osteocyte morphology. Results are quantified in **g–h** where each data point represents the dendrite length or dendrite number from an individual osteocyte. Data are presented as mean values ±SEM. **** indicates $p < 0.0001$ for comparison between cells measured from controls and patients. **i** RT-qPCR in Ocy454 cells following overexpression of GFP, WT, and SP7$^{R316C}$ constructs. Each data point represents a biologically independent replicate. Data are presented as mean values ±SEM. Ordinary one-way ANOVA was used, exact $p$ values are 0.0023 for *Ostn* and 0.0015 for *Sost*.

**Histomorphometry**. Right tibiae from 8-week-old mice have been subjected to bone histomorphometric analysis (calcein labeling). All mice received IP calcein injections (Sigma-Aldrich, 20 mg kg$^{-1}$) at 2 days prior to sacrifice. The tibia was dissected and fixed in 70% ethanol for 3 days. Fixed bones were dehydrated in graded ethanol, then infiltrated and embedded in methyl methacrylate without demineralization. Undecalcified 5 μm longitudinal sections were obtained using a microtome (Leica Biosystems, RM2255). The observer was blinded to the experimental genotype at the time of measurement. Digital images were obtained via fluorescent microscopy.

**Silver staining of mouse osteocyte morphology**. As described previously[86], paraffin-embedded mouse tibia sections were deparaffinized and incubated in two parts, 50% silver nitrate and one part 1% formic acid in 2% gelatin solution for 55 min. Stained slides were then washed in 5% sodium thiosulfate for 10 min and subsequently dehydrated, cleared, and mounted. Quantification of the dendrite number and length were performed with ImageJ by thresholding grayscale images for dark, silver-stained lacunae and canaliculi.

**TUNEL staining**. Formalin-fixed paraffin-embedded decalcified tibia sections from 6-week-old and 8-week-old mice were obtained. For Terminal deoxynucleotidyl transferase dUTP nick end labeling (TUNEL), sections were fixed with 4% paraformaldehyde (PFA) (Pierce™, 28908) in PBS for 15 min at room temperature and permeabilized with PCR grade recombinant Proteinase K (Roche Applied Science, 3115887001) for 30 min at 37 °C. Apoptotic cells were examined with Roche in situ cell death detection kit (Roche Applied Science, 11684795910) according to instructions and followed by 4′,6-diamidino-2-phenylindole (DAPI; Invitrogen) for fluorescent microscopy.

**RNAscope**. Formalin-fixed paraffin-embedded decalcified tibia sections (5 μm) from 6-week-old mice were obtained. Bone sections were processed for RNA in situ detection using RNAscope 2.l5 HD Assay-Brown (Chromogenic) according to the manufacturer's instructions (Advanced Cell Diagnostics[87],). For antigen retrieval, bone sections were pretreated with hydrogen peroxide and pepsin (1 h at 40 °C, Sigma-Aldrich). Tissue sections were hybridized with target probes (2 h at 40 °C), amplified (Amp1–4: 40 °C; Amp5–6: room temperature) and chromogenic detected using DAB followed by counterstaining with hematoxylin (American Master Tech Scientific). RNAscope probes used were: *Ostn* (NM_198112.2, region 2–1144), *Tnc* (NM_011607.3, region 875–1830), *Kcnk2* (NM_001159850.1, region 734–1642), *Dpysl3* (NM_001291455.1, region 772–1884), *Fbln7* (NM_024237.4, region 198–1595) and *Dabp* (negative control). Representative figures are from $n = 3$ wild-type C57B6 mice.

**Micro-CT**. Femurs were harvested from 6-week and 8-week old mice after being fixed with 10% formalin for 1 day and stored in 70% ethanol. Micro-CT imaging was performed on a bench-top scanner (μCT 40, Scanco Medical AG, Brüttisellen, Switzerland) to measure the morphology of the femoral mid-diaphysis (Scanning parameters: 10 μm isotropic voxel, 70 kVp, 114 mA, 200 ms integration time). A 500 μm long region of the mid-diaphysis was scanned, bone segmented from surrounding tissue using a threshold of 700 mgHA cm$^{-3}$ and the geometry of the cortex analyzed using the Scanco Evaluation Program. Cortical porosity was measured as the total area of pores in the cortex divided by the total area of the cortex [cortical porosity (%) = (pore area/cortex area) × 100)].

**Third-harmonic generation imaging**. To image bone/interstitial fluid boundaries surrounding osteocytes in calvariae, *Sp7*$^{OcyKO}$ and control mice were anesthetized with isoflurane, and placed in a custom 3D printed heated mouse holder. An incision was made on the top of the skull. The skin was partly detached exposing the periosteum and the calvarial bone[23]. A microscope cover glass holder was used to hold a microscope cover glass over the dissected area, which was hydrated with PBS.

The imaging locations were consistently chosen in regions 3, 4, 5, and 6[88], 100–300 μm away from the coronal and central veins, while imaging depths were chosen, starting from ~10 μm below the bone surface. A turn-key laser with a 5 MHz repetition rate, 370 fs duration pulses, and 1550 nm wavelength (FLCPA-01CCNL41, Calmar) was coupled to a 60 cm polarization-maintaining large mode area single-mode photonic crystal fiber with a 40 μm diameter core (LMA40, NKT Photonics, Birkerød, Denmark) which was coupled to an in-house laser scanning microscope. The microscope consisted of a polygonal laser scanner (DT-36-290-025, Lincoln Laser, Phoenix, AZ) for the fast axis and a galvanometric scanning mirror for the slow axis (6240H, Cambridge Technology, Bedford, MA), achieving 15 frames per second with 500 × 500 pixels. An oil-immersion objective lens (1.05 NA, UPLSAPO 30×SIR, Olympus, Waltham, MA) was used for imaging. Mineral oil (Sigma-Aldrich) was used as the immersion oil due to its high transmission at 1550 nm. THG imaging required 10 nJ of 1550 nm. Three-dimensional analysis of osteocytes was performed in ImageJ 1.50i (NIH, Bethesda, MD) and Imaris 7.4.2 (Bitplane Inc., South Windsor, CT).

**Phalloidin staining**. After growth in 2D or 3D, cells were washed with cold PBS after removing the culture medium. Cells were fixed with 4% PFA (Pierce™, 28908) in PBS for 10 min and permeabilized with 0.05% Saponin (Boston Bio-Products, BM-688) in PBS for 5 min. Cells were then incubated with phalloidin mixture (Abcam, ab176753, 1:1000) in the dark for 1 h at room temperature with gentle agitation. Cells were washed with cold PBS three times before being stained with DAPI (Invitrogen™) for 15 min. Cells were mounted with Fluoromount-G® (SouthernBiotech) and imaged by confocal microscopy. For 10 μm tibia cryosections, slides were first washed in cold PBS for 5 min. The following steps were the same as in culture cells. Quantification of osteocyte filament density was performed in ImageJ imaging software on a blinded basis. First, cell bodies were cropped, and then multi-color images were converted to single-channel (grayscale) color images. Each image was duplicated, and a binary image was created from the copied image. In the copied image, all dendrites were highlighted, and the background was subtracted. Next, we used "Analyze – Set Measurements" and set "Redirect to" to the original grayscale image, followed by selecting the "Area" and "Mean gray value" functions. Finally, the "Analyze – Analyze Particles" function was used to quantify the filament density.

**Cell viability, Annexin V, and EdU assays**. Ocy454 and MC3T3-E1 cells were cultured in 96-well plates at 37 °C for 7 days. From Day 1 to 7, cells were incubated each day for 1 h at 37 °C with 10% PrestoBlue solution (PrestoBlue™ Viability Reagent, Invitrogen™) containing culture media (alpha-MEM supplemented with heat-inactivated 10% fetal bovine serum and 1% antibiotic–antimycotic, Gibco™). Absorbance was read at 570 nm (resavurin-based color change) and 600 nm (background) to calculate cell viability per the instructions of the manufacturer. To detect apoptosis in shSp7 and shLacZ cells, we used fluorescein isothiocyanate (FITC) Annexin V Apoptosis Detection Kit with PI (BioLegend, 640914). We followed the manufacturer's instructions by staining cells with FITC Annexin V and Propidium Iodide Solution for 15 min at room temperature in the dark. Cell apoptosis was evaluated by flow cytometry. For EdU proliferation assays, cells were labeled with 5-ethynyl-2'-deoxyuridine followed by detection by click chemistry and then flow cytometry-based on the instructions of the manufacturer (ThermoFisher, C01425). All flow cytometry data were analyzed using FlowJo version 10.

**Flow cytometry and cell sorting**. For cells that were stained with phalloidin (Abcam, ab176753), cell pellets were resuspended with 2% FACS buffer (PBS plus 2% heat-inactivated fetal bovine serum) and then filtered through round-bottom tubes with cell strainer cap (Falcon®, 70 μm). For cells stained with Annexin V and EdU, cells were filtered through round-bottom tubes with a cell strainer cap (Falcon®, 70 μm). Flow cytometry was performed on a BD Sorp 8 Laser LSR II. For the enrichment of viable osteoblasts and osteocytes, cells were filtered through a 100 μm cell strainer to make a single-cell suspension. DAPI (Invitrogen™) was

added to cells before sorting. Dead cells, debris, doublets, and triplets were excluded by FSC, SSC, and DAPI. Viable tdTomato+/DAPI− cells were enriched on Sony SH800s Cell Sorter.

**Dual-luciferase reporter assay**. HEK293T cells were plated in 12-well plates at the density of 50,000 cells ml$^{-1}$. When confluent, cells were transfected using Fugene-HD (Promega) with a combination of (1) pGL4 *Firefly* luciferase reporter (Promega, E6651), (2) pRL *Renilla* luciferase control reporter (Promega, E2241), and (3) FLAG-hSP7 or FLAG-hSP7$^{R316C}$ or GFP control (VectorBuilder, EF1A as a promoter, vector IDs are VB190819-1114pvu, VB190819-1115xzu, and VB180924-1105fst) plasmids. Ostn_En2 sequence was synthesized with gBlocks Gene Fragments (IDT) (See sequence information in Supplementary Data 10). SacI and BglII restriction enzymes (NEB R0156S, R0144S) were used to cut both the pGL4 plasmid and the Ostn_En2 fragment. In all, 48 h later, cells were disassociated by adding Passive Lysis Buffer (Promega, E1910) and were gently rocking on an orbital shaker for 15 min. In all, 20 μl cell lysate was transferred to 96-well black polystyrene microplates (Corning). In all, 50 μl of Luciferase Assay Reagent II was dispensed to each well with the reagent injector. The sample plate was placed in the luminometer to measure the *Firefly* luciferase signal. In all, 50 μl of Stop & Glo Reagent was then dispensed to each well. The plate was placed back to the luminometer to measure *Renilla* luciferase activity. Luciferase experiments were performed in biologic triplicate, and all experiments were repeated at least twice.

**RNA isolation and qRT-PCR**. Total RNA was isolated from two-dimensional cultured cells using QIAshredder (QIAGEN) and PureLink RNA mini kit (Invitrogen$^{TM}$) following the manufacturer's instructions. In brief, lysis buffer with 2-mercaptoethanol was added to cold PBS washed cells and collected into QIAshredder columns, then centrifuged at 15,000 × *g* for 3 min. The flow-through was collected into a new tube and RNA isolation was carried out with PureLink RNA mini kit following the manufacturer's instructions. For three-dimensional cultured cells, cells were first treated with collagenase at 37 °C for 5 min before isolation using the PureLink RNA mini kit. For liver RNA isolation from 6-week-old mice, RNA was extracted by tissue blender with TRIzol (Life technologies) following the manufacturer's instruction, and further purification was performed with PureLink RNA mini column. cDNA was prepared with 1 μg RNA and synthesized using the Primescript RT kit (Takara Inc.). qPCR assays were performed on the StepOnePlus$^{TM}$ Real-time PCR System (Applied Biosystems) using SYBR Green FastMix ROX (Quanta bio). *β*-actin was used as the internal control for normalization. The $2^{-\Delta\Delta Ct}$ method was used to detect expression fold change for each target gene with three biological replicates. Supplementary Data 10 lists all primer sequences used.

**RNA-seq analysis**. RNA-sequencing was conducted using the BGISEQ500 platform (BGI, China)[89]. In brief, RNA samples with RIN values >8.0 were used for downstream library construction. mRNAs were isolated by PAGE, followed by adaptor ligation and RT with PCR amplification. PCR products were again purified by PAGE and dissolved in EB solution. Double-stranded PCR products were heat-denatured and circularized by the splint oligo sequence. The ssCir DNA was formatted as the final sequencing library and validated on bioanalyzer (Agilent 2100) prior to sequencing. The library was amplified with phi29 to general DNA nanoballs (DNBs), which were loaded into the patterned nanoarray followed by SE50 sequencing. We obtained at least 20 million uniquely mapped reads per library sequenced. *N* = 2–3 biologic replicates were performed for each condition. Sequencing reads were mapped to the mouse reference genome (mm10/GRCm38) using STAR v.2.7.2b[90]. Gene expression counts were calculated using HTSeq v.0.9.1[91] based on the Ensembl annotation file for mm10/GRCm38 (release 75). For the in vitro Ostn-rescue RNA-seq data sets, genes with expression counts of lower than 0.25 in less than two samples were removed. Differential expression analysis was performed using EgdeR[92] package based on the criteria of more than two-fold change in expression value versus control and false discovery rates (FDR) < 0.05. For *Sp7* knockdown and overexpression RNA-seq data sets, we first set the parameters at $\log_2 FC > 1$ and FDR < 0.05 to identify genes with Sp7-dependent expression. We then lowered the threshold to only FDR < 0.05 (eliminating the $\log_2 FC$ cutoff) to identify genes that were counter-regulated by Sp7 for cross-referencing to Sp7 ChIP-seq data sets.

Volcano plots and heatmaps were made using EnhancedVolcano and ggplot2/tidyverse packages from Bioconductor and tidyverse (https://github.com/kevinblighe/EnhancedVolcano, https://ggplot2.tidyverse.org). Gene Ontology enrichment analysis was performed with Metascape[93] and clusterProfiler[94]. Scatter plots were made in GraphPad Prism 8.0. The degree of differential expression overlap between two transcriptomic profiles was determined by Rank-Rank Hypergeometric Overlap (RRHO and RRHO2)[95,96]. Heatmaps generated using RRHO2 have the top right (both increasing) and bottom-left (both decreasing) quadrants, representing the concordant changes, while the top left and bottom right represent discordant overlap (opposite directional overlap between data sets). For each comparison, one-sided enrichment tests were used on $-\log_{10}$(nominal *p* values) with the default step size of 200 for each quadrant, and corrected Benjamini–Yekutieli *p* values were calculated.

**ChIP-seq**. ChIP was performed as previously described with minor modifications[30,97]. Protein–DNA complexes were crosslinked with 1% formaldehyde; the crosslinking was quenched with 250 mM glycine. Tissues were sheared by sonication to generate a chromatin size range of 200–600 bp. Dynabeads (sheep anti-mouse IgG,11201D Life Technologies) were pre-incubated with anti-FLAG M2 antibody (F1804, Sigma-Aldrich) incubating overnight with 500 μg of chromatin. Protein–DNA crosslinks were reversed by incubating at 65 °C overnight followed by RNase and Proteinase K treatment. Samples were purified with MinElute PCR purification kit (Qiagen, 28804). The construction of ChIP-seq libraries was performed with a ThruPLEX®-FD Prep Kit (R40012, Rubicon Genomics) according to the manufacturer's instruction. The library was sequenced on HiSeq X (Illumina) platforms.

Sequencing read quality was evaluated using FastQC (http://www.bioinformatics.babraham.ac.uk/projects/fastqc/). For both osteocyte and POB data sets, ChIP-seq reads were aligned to the mouse genome (mm9) using Bowtie2[98] with the parameters described previously[99]. Sp7 peaks were identified using MACS2[100] with default parameters except for the effective genome size, which was set for a mouse (mm9). The intersection between osteocyte Sp7 peaks and POB Sp7 peaks were performed with BEDTools[101]. The genomic annotation was assigned to Sp7 peaks using ChIPseeker[102]. Peaks were associated with Gene Ontology (GO) terms using the Genomic Regions Enrichment of Annotation Tools (GREAT)[103]. The assignment of target genes was performed by associating Sp7 peaks with neighboring genes using GREAT. De novo motif enrichment was performed within ±50 bps of Sp7 peak summits using DREME[32]. BigWig files from H3K27ac, H3K4me, and H3K4me3 ChIP-seq experiments in IDG-SW3 cells were downloaded from the NCBI GEO database (GSE54784[31]). The intensity of histone modifications at Ocy-specific Sp7 peaks was examined by deepTools[104].

**Cell painting**. Control (shLacZ) and shSp7 Ocy454 and MC3T3-E1 cells were grown in 384-well plates. Cells were treated with 100 nM CNP, 500 nM OSTN, or both, for 48 h prior to fixation and staining according to previously published procedure[105], with some adjustments for the final dye concentrations: MitoTracker Deep Red (Invitrogen M22426, 0.5 mM), Alexa Fluor 568 phalloidin (Invitrogen A12380, 8.25 nM), Wheat Germ Agglutinin, Alexa Fluor 555 Conjugate (Invitrogen W32464, 0.0015 mg/mL), Concanavalin A, Alexa Fluor 488 Conjugate (Invitrogen C11252, 0.005 mg/mL), SYTO 14 green-fluorescent nucleic acid stain (Invitrogen S7576, 6uM), Hoechst 33342 trihydrochloride trihydrate (Invitrogen H3570, 0.001 mg/mL). Imaging was done using PerkinElmer Opera Phenix instrument; non-confocal mode; ×20 water objective; nine fields per well. We extracted morphology features from the Cell Painting images using CellProfiler-4.1.3 pipelines[106]. We first performed illumination correction for each channel on a per-plate basis. Next, we segmented cells, identified nuclei and cytoplasm, and then measured specific features for each of the captured channels from the illumination-corrected images. We measured the fluorescence intensity, size, shape, colocalization, granularity, texture, and several other measurements for each cell (see https://cellprofiler-manual.s3.amazonaws.com/CellProfiler-4.1.3/index.html for more details). Following the image analysis pipeline, we obtained 5962 feature measurements for the MC3T3-E1 and Ocy454 cell lines. We next used a standard approach[107] to process the single-cell profiles. First, we aggregated all single cells per well by computing a mean value per morphology feature. Next, we normalized the feature distributions separately for each individual cell type plus genotype to facilitate seeing the effects of individual treatments in each background. We then applied a feature selection procedure to remove features with missing values in any profile, low variance, and extreme outliers. The Granularity_14, Granularity_15, Granularity_16, Manders, Costes, and RWC measurements were removed manually, since they are known to be noisy and have interfered with previous analyses. We used cytominer (https://github.com/cytomining/cytominer) to perform the profiling pipeline. Following these procedures, 1037 morphology features remained, and these were further analyzed within the Morpheus platform (Broad Institute). Similarity matrices (Pearson correlation) for each cell type/genotype were calculated after median-collapsing six replicate wells per condition and include all 1037 features. Marker selection was then used on non-collapsed replicates to identify specific features affected by CNP or by OSTN.

**Serial digestion and single-cell RNA-seq**. Four-week-old mice (*n* = 2 per genotype) were sacrificed and tibiae, femura, and calvariae were dissected. Soft tissue was removed through scraping, and the epiphysis was cutoff. Bone marrow cells were flushed out with cold PBS using a syringe. Bones were cut into 1- to 2-mm lengths and subjected to eight serial digestions as described previously[108,109]. In short, bone pieces were incubated in the sequence of 15 min collagenase (0.2% of collagenase type I in isolation buffer, Worthington)/15 min EDTA solution (5 mM EDTA, 0.1% BSA in PBS)/15 min collagenase solution/15 min EDTA solution/15 min collagenase solution/30 min EDTA solution/30 min collagenase solution/30 min collagenase solution. Collagenase and EDTA solutions were prepared with RNAse inhibitor added (1:100 ratio, Lucigen, 30281). Isolation buffer is composed of 70 mM NaCl, 10 mM NaHCO$_3$, 60 mM sorbitol, 30 mM KCl, 3 mM K$_2$HPO$_4$, 1 mM CaCl$_2$, 0.1% BSA, 0.5% glucose and 25 mM HEPES. Each digestion took place in 5 ml solution in a 15 ml centrifuge tube on the thermomixer set at 35 °C/500 rpm. Bone fragments were washed in PBS between digestions. The final three collagenase fractions were collected by centrifuging the supernatant at 4 °C 300 rcf

for 8 min. Cell pellets were resuspended with 2% FACS buffer containing RNase inhibitor and were filtered through a 100 μm cell strainer.

DAPI (Invitrogen™) was added to cells before sorting. Dead cells, debris, doublets, and triplets were excluded by FSC, SSC, and DAPI. tdTomato+/DAPI− cells were sorted (Sony SH800s Cell Sorter) into a new PCR tube containing 2% FACS buffer. Single cells were encapsulated into emulsion droplets using Chromium Controller (10x Genomics). scRNA-seq libraries were constructed using Chromium Single-Cell 3' v3 Reagent Kit according to the manufacturer's protocol. In brief, ~15,000 cells were loaded in each channel with a target output of 2,000 cells. Reverse transcription and library preparation were performed on C1000 Touch Thermal cycler with 96-Deep Well Reaction Module (Bio-Rad). Amplified cDNA and final libraries were evaluated on an Agilent BioAnalyzer using a High Sensitivity DNA Kit (Agilent Technologies). Libraries were sequenced with PE100 on the DNBSEQ™ NGS technology platform (BGI, China) to reach ~ 450 million reads.

**Single-cell sequencing read processing and analysis**. Raw reads obtained from scRNA-seq experiments were demultiplexed, aligned to the mouse genome, version mm10 (with tdTomato gene inserted), and collapsed into unique molecular identifiers (UMIs) with the Cellranger toolkit (version 3.1.0, 10X Genomics). After generating digital count matrices using the count function in CellRanger, libraries were batch-corrected to achieve a similar average read count using the CellRanger aggr function. Using a cutoff of 1000 UMIs, we then performed cell-type annotation iteratively through two rounds of dimensionality reduction, clustering, and removal of putative doublets across all cells. For the first level clustering, we used a modified workflow of LIGER, using integrative non-negative matrix factorization (iNMF) to limit any experiment-associated batching. In brief, we normalized each cell by the number of UMIs, selected highly variable genes, performed iNMF across both experimental conditions (WT and $Sp7^{OcyKO}$), and clustered using Louvain community detection, omitting the quantile normalization step. Visualizations were obtained and downstream trajectory algorithms were carried out on the lower-dimension embedding obtained by UMAP (Uniform Manifold Approximation and Projection).

To test for significance of proportions across all major osteoblast/osteocyte subtypes, high-quality (low percentage of mitochondrial reads) cells were used to test for differences in proportions. A two-by-two contingency table was constructed iteratively per cell type by summing all cells within and outside a defined cluster per genotype. A Barnard's exact test was run using the Exact package in R.

Differential expressions for all the major cell types were performed using a Wilcoxon rank-sum test from the presto package. We removed cells with a high expression of mitochondrial genes (percentage of reads mapped to mitochondrial reads greater than 10%), removing 850 cells from the WT library and 506 from the $Sp7^{OcyKO}$ library. All non-osteoblast/osteocyte populations as defined by the first-round annotation were then excluded, and the modified workflow was run again to define all subpopulations. Top marker genes for all subpopulations of osteoblast/osteocytes were obtained from a Wilcoxon rank-sum across all cells from the control library. The ratio of the number of osteoblast/osteocyte subtypes was determined per library by taking the number of cells within the cluster and dividing by the total number of cells obtained per library (to control for experimental differences in the number of cells captured).

Differential expression between $Sp7^{OcyKO}$ and WT was carried out across all osteoblast/osteocyte subtypes using a Wilcoxon rank-sum test with an equal number of cells per library to ensure results were not confounded by relative changes in the number of cells with the subtype.

**Trajectory analyses—Monocle3 and velocyto**. Pseudotime trajectory inference was carried out using the workflow suggested in the Monocle3 tutorial (http://cole-trapnell-lab.github.io/monocle-release/monocle3/#tutorial-1-learning-trajectories-with-monocle-3). In brief, UMAP projections obtained for the osteoblast/osteocyte subtype analyses were used to learn a graph of cell connectivity. A root node was then defined by the most connected vertex within the Canonical Osteoblast ($Bglap3$) subtype. The pseudotime values were then displayed on the UMAP projection.

For the velocyto analyses, spliced and unspliced matrices were generated by executing the run command from the velocyto package on both libraries. Spliced and unspliced loom files were then read in using the velocyto.R package. Genes were filtered based on their expression value across all subtypes by setting the min.max.cluster.average = 0.5 for spliced and 0.05 for unspliced matrices for the control data set and 0.8 and 0.08 for the spliced and unspliced matrices in the $Sp7^{OcyKO}$ data set, respectively. The cell distance matrix was determined by creating a correlation matrix of the factors generated by the iNMF computation from LIGER. Velocities were determined using the gene-relative slopes with the following parameters (deltaT = 1, kcells = 25). Finally, velocity arrows were displayed on the UMAP embedding generated from the LIGER clustering with the following parameters (neighborhood size = 100 cells, velocity scale = "sqrt" (square root), minimal cell "mass" (weighted number of cells) around each point = 0.5, number of grid points = 40).

**Heritability enrichment of cell types for human fracture risk**. MAGMA (Multi-marker Analysis of GenoMic Annotation) gene annotations were carried out by running the annot command-line tool across all SNPs from the hg19 human genome build. Summary statistics from fracture risk were downloaded from the Genetic Factors for Osteoporosis (GEFOS) consortium website (http://www.gefos.org/?q=content/ukbb-ebmd-gwas-data-release-2017). A z score for all genes in the hg19 annotation was then obtained by aggregation of SNP p values using the default settings in the MAGMA command-line tools.

To test the significance of associated enrichment of fracture risk in marker genes across all major cell types, we first ran a Wilcoxon rank-sum test on all major cell types from our first round of LIGER-based clustering (aggregating all osteoblast/osteocyte subtypes together) using only cells from the wild-type mouse library. After converting mouse genes to human ones using the homolog file downloaded from the Jackson Labs website (http://www.informatics.jax.org/faq/ORTH_dload.shtml), we gathered all genes per cell type with a log-fold change greater than 0.02. The gene z scores defined by MAGMA were then regressed against the log-fold change values per major cell type using default parameter settings from the MAGMA gene-set analysis. Bonferroni-corrected p values and effect sizes (BETA values) were obtained for each major cell type, corresponding to the degree of enrichment of genes associated with fracture risk within the marker genes for each cell type.

**Identification of cell-type-specific Sp7 target genes**. In all, 10,148 Sp7-bound regions were identified as osteocyte-specific (Ocy-specific) enhancers from Sp7 ChIP-seq analysis and 6648 genes were associated with these enhancer regions. 146 genes were counter-regulated by $Sp7$ knockdown and overexpression from bulk RNA-seq analysis. In all, 77 genes were identified as Ocy-specific Sp7 target genes when intersecting two lists of genes (6648 and 146). Similarly, 1733 POB-specific enhancers were identified and 1962 genes were strongly associated with these enhancer regions. We then intersected this list of genes to the POB-specific genes derived from RNA-seq comparison between osteoblasts and chondrocytes[30], and identified 134 POB-specific Sp7 target genes.

**Sp7 target gene-set enrichment analysis in DropViz data set**. To determine the degree of enrichment of cell type-specific Sp7-associated genes within the major cell types in the murine brain, a custom enrichment score was created using an empirical null distribution of gene sets from the DropViz data set (PMID: 30096299). First, summed gene values for all transcriptionally-defined cell types were downloaded from the DropViz website (http://dropviz.org/). A Wilcoxon rank-sum test on the aggregated expression values was carried out between all major cell types in the mouse brain (astrocytes, choroid plexus cells, endothelial/mural cells, ependymal cells, microglia/macrophages, mitotic cells, neurogenesis-associated cells, neurons, and oligodendrocytes/oligo precursor cells) to define average expression differences for all genes for the major cell types. To generate a null distribution of gene-set enrichment values, per cell type, the average mean difference was calculated on a randomly sampled set of 77 genes. This procedure was iterated 500 times per cell type and an empirically defined p value was then calculated by comparing the number of gene sets with a greater average enrichment value for that cell type than that of the Sp7-associated 77 genes, corresponding to the cell type-specific degree of enrichment for the Sp7-associated gene-set. Of note, similar enrichment patterns were obtained using linear mean expression difference and fold change methods (Supplementary Fig. 17a). This same procedure was also performed on the 134 POB-specific Sp7 targets and the top 150 (ranked by p value) marker genes of each cluster derived from the mouse bone scRNA-seq data set (Fig. 6o–p).

**Association of mature osteocyte markers with skeletal dysplasia**. Mature osteocyte marker orthologs were identified in the Nosology and Classification of Genetic Skeletal Diseases[54]. Top400 mature osteocyte markers were derived from the markers identified from cluster 6 of WT single-cell profile based on a percentage of difference. Significant enrichment of mature osteocyte marker orthologs among all genes in the nosology, and within each of the skeletal dysplasia groups was calculated using RStudio (http://www.rstudio.com, V1.2.5033). The network plot was constructed using Cytoscape[110]. Each gene was colored based on a percentage of difference and enriched disease groups were colored according to p value.

**Quantification and statistical analysis**. All experiments were performed at least twice. Data are expressed as means of triplicate biological repeats within a representative experiment plus/minus standard error. Statistical analyses between two groups were performed using an unpaired two-tailed Student's t test (Microsoft Excel). When more than two experimental groups were present, ANOVA analysis followed by post hoc Tukey–Kramer test was performed. P values <0.05 were considered to be significant. Variation between groups was similar in all cases.

**Reporting summary**. Further information on research design is available in the Nature Research Reporting Summary linked to this article.

## Data availability

The RNA-seq, ChIP-seq, and scRNA-seq data are deposited in NCBI's Gene Expression Omnibus (GEO) "GSE154719". The authors declare that all other data supporting the findings of this study are available within the article and its supplementary information files or from the corresponding author upon reasonable request. Source data are provided with this paper.

## Materials availability

Unique materials and reagents generated in this study are available upon request from the lead contact.

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

## Acknowledgements

We thank Drs. Tatsuya Kobayashi, Amar Sahay, Jay Rajagopal, Lauren Surface, Ryan Logan, Marianne Seney, and all members of the Wein laboratory for discussions. We thank Dr. Benoit de Crombrugghe for providing *Sp7* conditional knockout mice, Dr. Vanda Jorgeti for providing healthy age-matched control iliac crest samples, and Dr. Frank Rauch for providing iliac crest biopsy samples from *Sp7*<sup>R316C</sup> mutant patients. M.N.W. acknowledges funding support from the MGH Department of Medicine (Transformative Scholars award), the American Society of Bone and Mineral Research (Rising Star Award), and the National Institute of Health (AR067285). C.M.M. acknowledges support from the NIH (T32DK007028). M.N.W. and C.M.M. acknowledge generous support from Louise Pearl Corman, Ph.D. μCT and bone histomorphometry were performed by the Center for Skeletal Research, an NIH-funded program (P30 AR066261). Confocal microscopy was supported by the NIH Shared Instrumentation Grant (SIG) S10OD021577.

## Author contributions

J.S.W., C.M.M., F.M., D.R., H.H., C.D.C., N.T., R.P., N.G., T.E., Y.W., Jd.S.M., M.B., D.J.B., D.T., A.A., and M.N.W. conceived the study design and performed experiments. J.S.W., T.K., F.M., H.H., M.L.B., E.Z.M., P.R., B.C., M.K.A., P.B., D.T., C.P.L., H.M.K., and M.N.W. analyzed data. M.F., C.F.M., and M.F. contributed key reagents. J.S.W. and M.N.W. wrote the manuscript. All authors edited and approved the manuscript.

## Competing interests

The following authors declare the following competing interests: M.N.W. and H.M.K. receive research funding from Radius Health and Galapagos N.V. All other authors declare no competing interests.

## Additional information

**Peer review information** *Nature Communications* thanks Matthew Greenblatt and the other anonymous reviewer(s) for their contribution to the peer review this work. Peer reviewer reports are available.

