## [Peer Review File · Nature Communications]

Reviewers' Comments:

Reviewer #1:

Remarks to the Author:

In the current manuscript, Wang et al set out to identify gene regulatory networks that control osteocytogenesis. The authors identify Sp7 as a key regulator of osteocyte dendritogenesis and characterize osteocrin as a downstream target required for dendritogenesis. The authors also perform single cell analysis to characterize the various subpopulations of cells that are part of the osteoblast – osteocyte differentiation process. Overall, this is a well conducted study that adds significant new information about the processes that control osteocyte formation. Osteocytes are deeply embedded in bones, and it is challenging to study them. The authors developed a 3D culturing system to study osteocytes in vitro, and nicely complement their studies with in vivo experiments.

Major comment:

The phenotypic and molecular case for Sp7 requirement for dendrite formation in osteocytes is strong and robust. The correction by osteocrin is potentially quite novel. It is unfortunate the mechanistic basis of this dramatic effect was unstudied. The authors have tried to keep their manuscript short and to the point. However, their brevity goes at the cost of experimental clarity. Many figure panels are barely described, and some are not even mentioned in the main text at all. I highly recommend the authors to rework their manuscript to make sure that all experiments are properly described and interpreted. In addition, the figures should be reorganized to follow the flow of the text. As they are now, the reader has to go jump between figures to follow the text.

Minor comments:

- 1) The combined expression of Atf3, Klf4, Pax4 and Sp7 induces ectopic Sost expression in human fibroblasts. The authors mention that in their hands, only Sp7 induced ectopic sost expression in Ocy454 cells. The authors do not mention whether they tried any combinations of the various transcription factors, or only expressed one transcription factor at a time. On a similar note, later in the manuscript (line 161), the authors identify the binding site of the likely cofactor of Sp7. Is any of the three remaining transcription factors (Atf3, Klf4 or Pax4) predicted to use this sequence as a binding site?
- 2) Fig 2 & Fig S2: the authors switch back and forth between OCY454 and MC3T3 cells and it is not always clear which cell line was used for which experiment. It appears that they have performed all analyses on both cell lines and obtained consistent results, but please specify this better in the main text rather than in the figure legend.
- 3) Suppl figure 1E is missing a panel displaying GFP to show expression of the control vector.
- 4) What caused the increased intracortical bone formation and increased bone resorption in Sp7OcyKO mice? The authors should test bone turnover markers.
- 5) Fig 4G: Could the authors provide the list of 77 identified genes as an actual table embedded in the supplemental figures? At the moment, it is buried within an excel sheet with multiple tabs.
- 6) Fig 5: The rescue of the osteocyte phenotype using liver expressed osteocrin is quite impressive. Given the function of osteocrin as a regulator of naturetic peptide signaling, were there effects in bone growth, growth plate, bone mass? Do the authors have a proposed mechanism for this dramatic rescue?
- 7) Fig S3B: Only one of the three constructs used to knock down Ostn expression manages to alter the phalloidin spectrum. Could the authors show the knockdown efficiency of all three constructs to show that this is not due to an off-target effect of the 3rd construct used?
- 8) Fig S3D: Please find correlating images to show loss of Ostn expression in Sp7 cKO mice. The periosteum does not appear to be shown in the bottom panel.
- 9) Fig S4B: If loss of Sp7 alters dendrites and changes phalloidin staining, actin is not a good reference gene to normalize expression levels to.
- 10) S4E: Please show Ostn-expression in bone, rather than liver tissue.
- 11) The human data shown in this manuscript is quite interesting. The authors hypothesize that the OI-related mutation in Sp7 is due to a loss of Sp7-function in osteocytes. If this is true, one would expect to find reduced Ostn expression in the human bone samples shown in Fig 7E, while expression of the osteoblast-specific sp7 target genes identified earlier in the paper would not

change. Alternatively, could the authors show that overexpression of WT versus mutant Sp7 in vitro does not have an altered ability to induce expression of Ob-specific genes?

Reviewer #2:

Remarks to the Author:

This is exciting work showing the potential role of Sp7 and osteocrin in dendrite formation. The rescue of the Sp7 deletion bone defect by osteocrin is impressive. The most novel aspect of this study is the single cell sorting which identified several clusters including one with high expression of *Sost*, therefore an osteocyte cluster. Another novel finding is that osteocyte specific genes are targets of Sp7 and distinct from osteoblasts and share similarity with neurons.

However, there are some concerns.

It is very hard to prove that a change in osteocyte dendrite formation is responsible for a bone phenotype or disease. It can only be inferred that a defect identified in dendrite formation in in vitro experiments translate into the same defect in vivo. The bone defect could also be due to other effects of Sp7 such as on matrix formation.

Fig 1. Increased porosity suggests an increased in osteoclast activity. Greater osteocyte cell death is observed suggesting an increase in Rankl. It is postulated that a reduction in dendrites causes the increase in osteocyte cell death. It would be important to determine if Rankl is elevated in osteocytes in the Sp7 targeted deletion mice.

Fig. 2. Normal osteocytes do not show 'spines' on their dendrites. Therefore it is suggested that the dendrites of MC3T3 cells do not represent osteocytes. Do the Ocy454 also show 'spines' on their dendrites? If not, these are a more representative model of primary osteocytes.

The actual number of dendrites per cell should be counted. The image in 2a is qualitative and not quantitative. For Image 2b, a decrease in phalloidin expression would not necessarily mean a decrease in dendrite number/cell. Again, the actual number of dendrites/cell should be quantitated.

It is stated that MC3T3 cells grown in 3D culture have changes consistent with osteocyte maturation (SFig2c and STable 2). What are these changes?

Fig 3. The authors have examined genes regulated by knockdown and over expression of Sp7 in Ocy454 cells. They focus on neuronal genes but ignore the others such as ossification, matrix organization, etc. Just because a pathway is identified does not necessarily mean the pathway is performing the same function in a different cell type, for example, though genes involved in brown fat and inner ear development are also regulated in the Ocy454, that does not mean they have the same function in osteocytes.

77 direct Sp7 target genes were identified in osteocytes and a significant number of neural genes were expressed. This figure should go into the main text instead of in the supplementary data (Fig S7). This data supports the author's hypothesis.

A clear explanation for why osteocrin was chosen over other Sp7 target genes was not provided.

The authors provided no rationale for why they decided to choose this particular target for performing immunostaining of the Sp7 mutant bones. Was this fishing or was there a reason?

Fig 6. How was each cluster annotated? Why were the different genes chosen such as *Pdgfr1*, *Mgst3*, etc. What about other markers for different stages of osteoblast to osteocyte differentiation?

Supp Fig S6: It would be important to show the expression of additional osteocyte genes such as *Mepe*, *Fgf23*, etc. in addition to the mature osteocyte marker *Sost*, and early osteocyte markers, *Dmp1*, *Pdpr*, and the late osteoblast gene, *Bglap3*. Where is the data showing that *Ptprz1* is a terminal osteocyte marker?

1). With regards to the single cell experiments, why did the authors include calvaria in their bone samples? It is very difficult to remove marrow from calvaria with collagenase digestion/chelation. Could the calvaria bone marrow have skewed the single cell results?

2). Why are the authors using MC3T3-E1 cells when the Ocy454 have much higher levels of Sp7 and there is complete knockdown in the Ocy454 but not the MC3T3 (supplementary figure S1)? Concerns are expressed above about abnormal dendrites, etc.

Reviewer #3:

Remarks to the Author:

In this manuscript, Wang et al. identify that osterix/Sp7 has an unexpected role to regulate osteocyte dendrogenesis, at least in part through osteocrine as a downstream secreted effector. Overall, this work has several notable strengths that result in a high level of enthusiasm and high likely impact. First, this work establishes a new and unexpected function for Sp7 and takes important steps beyond existing literature towards establishing osteocyte dendrogenesis as a distinct biologic processes required for maintenance of cortical bone strength. This work also correspondingly establishes systems where osteocyte dendrogenesis can be studied in vivo and in vitro, therefore opening this area to further study. This work also provides an explanation for observations in the literature that several genes likely play fundamentally very distinct roles during the osteoblast to osteocyte transition, than they do during their better studied functions during osteogenic lineage commitment. This report provides an example of how even well studied factors involved with the early stages of osteoblast differentiation can play a distinct dual purpose in later osteocyte transition, by those factors engaging a distinct set of transcriptional partners and by mediating distinct processes, such as osteocyte dendrogenesis, that don't occur during early osteoblast differentiation. Lastly, it is notable that convincing data is presented that the mechanisms identified here operate in humans. Overall, the experiments appear to be technically sound and a large amount of data is provided in support of the conclusions. There are a few important questions, largely about the interpretation of data, that should be resolved.

Major points: 1. This work seeks to draw comparisons between neural dendrogenesis and osteocytic dendrogenesis. Is there a role for osterix in neurons? If feasible, a basic evaluation of *Osx*/Sp7 function in neurons would be helpful in judging how universal this Sp7 pathway is to dendrogenesis. Even if negative, this data would be of interest for clarifying that the convergence between neuronal and osteocytic dendrogenesis happens at the level of a shared set of target gene effectors, despite distinct transcriptional mechanisms being used to engage that dendrogenesis program in each tissue.

2. Some clarification and additional data is needed regarding the uCT studies. What methods were used to measure cortical porosity? Specifically, are the values reported in Fig 1e percent of pore volume over total cortical volume or is some other volume used as the denominator? Is trabecular bone mass intact in these mice? Is there a calvarial phenotype? The calvarial phenotype is of interest given the use of the calvarial site for some of the studies characterizing dendritic morphology. Is the overall cortical diameter unchanged vs controls? Have these mice been observed to display spontaneous fractures?

Minor points

1. Generally the methodologic description of FACS and single cell sequencing analysis generally provides a good degree of detail, but a few additional details would both help in judging this data and also facilitate reproduction. The FACS plots used for DAPI sorting and doublet exclusion should be provided in supplemental figures. Was filtering based on relative amounts of ribosomal or mitochondrial transcripts performed? While osteocytes are generally considered post-mitotic, it is possible that some of the less mature populations captured could be entering cell cycle. Do any of the clusters show strong differences in cell cycle indicating that this is driving the clustering? A similar question pertains to whether there are any clusters that show clear differences in the number of UMIs per cell equivalent, though it is appreciated that likely doublet filtering and some normalization steps were applied.

2. If feasible, in Fig 1g, if the morphologic parameters of the dendrites can be explored in further detail, this would enhance the data presentation. Can mean total dendrite length be distinguished and reported separately from the mean number of branch points each dendrite displays (perhaps with a normalization for dendrite length) using the harmonic imaging data? Is this a phenotype of mean dendrite length or a defect in branching or both? It would appear feasible to count the branch points manually if needed. As this area of research becomes more sophisticated, it seems likely that there will be phenotypes that separately influence dendrite extension and branching, so distinguishing these properties is of interest. Along similar lines, can osteocyte-osteocyte dendritic connections be counted in the harmonic imaging data, this would also be of interest to emphasize that not only is dendritic length reduced, but that this results in loss of functional contacts among osteocytes.

3. Similar to the comment above, it would strengthen the presentation if density of both lacunae and osteocytes per areal unit of bone cortex can be reported. Fig 1 makes a convincing case the

osteocytes die at an increased rate in the Sp1 dmp1-cre mice, resulting in a large number of empty lacunae. It then becomes of interest to understand whether the rate of osteocyte generation/the rate of osteoblasts proceeding to become matrix embedded osteocytes is impacted. Perhaps measuring the density of lacunae and then normalizing to the bone formation rate measured by histomorphometry would allow estimation of the rate of osteocyte generation? Clarity whether the rate of osteocyte generation is impacted will be useful to aid in interpreting the single cell seq data in Fig 6 in terms of understanding whether there is clear evidence for an osteoblast to osteocyte differentiation block in vivo.

4. As an optional suggestion, it may be of interest to highlight a few of the Sp7 dependent genes that are known to be or are plausibly involved in dendrogenesis. These could be distilled into a small figure or supplementary figure perhaps showing an expression heatmap with a few manually selected with the degree of change upon Sp7 loss and gain of function. This would also provide an opportunity to briefly highlight or discuss some relevant neural literature that establishes these genes as involved with neuronal dendrogenesis. This would help to further develop the claim that Sp7 is a regulator of dendrogenesis and will also likely be of interest to most readers.

5. *Ostn* is described as an osteocyte-specific transcript, but the data in Fig 3c suggests that it is expressed in what appear to be periosteal osteoblasts based on morphology. Clarification or, if needed, changes to terminology used to describe *Ostn* expressing cells would be helpful. If *Ostn* is expressed in osteoblasts, does this indicate that the cells receptive to *ostn* signals are later stage cells where dendrogenesis occurs and that *ostn* is not acting in an autocrine manner? Was *Ostn* expressed in the scRNA-seq data and, if so, which clusters displayed expression? Was this expression diminished in Sp7-deficient samples by scRNA-seq? It seems like there is likely a complex effect of Sp7 such that it has distinct impacts at multiple distinct steps in the osteocyte differentiation process. Exploring the existing data further from this perspective to start to make some of these issues more explicit in the manuscript would likely aid with clarifying these effects.

6. Related to the above point, the *Ostn* rescue is very impressive and has interesting potential therapeutic implications. However, there appears to be a disconnect between this *Ostn* rescue with the idea that Sp7 is directly promoting transcription of effector genes for dendrogenesis. Is it the case that the main Sp7-dependent effector is *Ostn* and the various dendrogenesis effector genes are only indirectly regulated by Sp7 via an *Ostn*-dependent autocrine loop? Or is it instead the case that both Sp7 and *Ostn* have additive and direct effects on the transcription of dendrogenesis effector genes? It is appreciated that it can be challenging to fully disentangle these possibilities. However, perhaps the Sp7 ChIP data combined with the various transcriptional profiling studies already performed can be used to at least partially clarify this question.

7. The finding of CAR cells in the DMP1-cre scRNA-seq studies is unexpected, as these cells are not normally near the cortical surface and are thus not often considered in relationship to osteocytes. This should likely be commented on in the manuscript.

8. Can any of the Sp7 regulated candidate effectors of dendrogenesis be detected in the scRNA-seq data? Does this provide insight into which cluster serves as the point where dendrogenesis is initiated? It would not be surprising if dendrogenesis is restricted to *SOST*+ classical mature osteocytes, but would be interesting if it is primed at an earlier stage of differentiation. Can differences in expression of these dendrogenesis effector genes also be appreciated in scRNA-seq studies of Sp7-deficient samples? This type of analysis may also be helpful for better understanding the relationship among the several processes occurring that are regulated by Sp7: 1) regulation of *ostn* expression, possibly in an osteoblast-like cell, 2) regulation of dendrogenesis, possibly through both *ostn*-dependent and *ostn*-independent by sp7-dependent mechanisms, 3) a differentiation blockade as suggested by the single cell seq clustering data, 4) osteocytic cell death in the absence of Sp7.

9. It appears that the demographic information for the patients contributing specimens is in reference 15? If available, it may be helpful to reproduce some of this information in the methods or corresponding figure legend to better clarify the characteristics of patients contributing specimens to the study.

Reviewer 1

In the current manuscript, Wang et al set out to identify gene regulatory networks that control osteocytogenesis. The authors identify Sp7 as a key regulator of osteocyte dendritogenesis and characterize osteocrin as a downstream target required for dendritogenesis. The authors also perform single cell analysis to characterize the various subpopulations of cells that are part of the osteoblast – osteocyte differentiation process. Overall, this is a well conducted study that adds significant new information about the processes that control osteocyte formation. Osteocytes are deeply embedded in bones, and it is challenging to study them. The authors developed a 3D culturing system to study osteocytes in vitro, and nicely complement their studies with in vivo experiments.

We thank the reviewer for these supportive comments.

Major comment:

The phenotypic and molecular case for Sp7 requirement for dendrite formation in osteocytes is strong and robust. The correction by osteocrin is potentially quite novel. It is unfortunate the mechanistic basis of this dramatic effect was unstudied. The authors have tried to keep their manuscript short and to the point. However, their brevity goes at the cost of experimental clarity. Many figure panels are barely described, and some are not even mentioned in the main text at all. I highly recommend the authors to rework their manuscript to make sure that all experiments are properly described and interpreted. In addition, the figures should be reorganized to follow the flow of the text. As they are now, the reader has to go jump between figures to follow the text.

We thank the reviewer for raising these points. As detailed below, we have now performed additional studies (**new Fig. 5m, Supplementary Fig. 7b, Supplementary Figure 10**) to define the mechanism through which osteocrin promotes osteocyte dendrite development and maintenance. In addition, **we have significantly revised the manuscript** throughout to ensure that all experiments are completely described and interpreted. Furthermore, we have significantly expanded the Discussion section to address the excellent points raised by all three reviewers.

Minor comments:

1) The combined expression of Atf3, Klf4, Pax4 and Sp7 induces ectopic *Sost* expression in human fibroblasts. The authors mention that in their hands, only Sp7 induced ectopic *sost* expression in Ocy454 cells. The authors do not mention whether they tried any combinations of the various transcription factors, or only expressed one transcription factor at a time.

On a similar note, later in the manuscript (line 161), the authors identify the binding site of the likely cofactor of Sp7. Is any of the three remaining transcription factors (Atf3, Klf4 or Pax4) predicted to use this sequence as a binding site?

This is a very interesting and important series of questions. As noted, combined ectopic expression of Atf3, Klf4, Pax4, and Sp7 in dermal fibroblasts induced *Sost* expression. The purpose of our study was to investigate the endogenous role of these transcription factors in osteocytes. For this reason, we started with loss of function studies (Supplemental Fig. 1a) for these 4 factors, and noted that reducing only Sp7 levels reduced *Sost* expression and sclerostin secretion. Subsequent over-expression studies (Supplementary Fig. 1b) further confirmed a role for Sp7 in sclerostin regulation. We did not test over-expression of the other transcription factors in the absence of clear loss of function data.

The second point embedded within this question is extremely interesting. From our Sp7 ChIP-seq performed in Ocy454 cells, we identified TGA(G/T)TCA as the top *de novo* motif of osteocyte-specific Sp7 enhancers. Atf3 is reported to recognize both TGACGTCA and TGA(G/C)TCA motifs.

Figure R1. 293T cells were transfected as indicated along with osteocrin -110 kB enhancer firefly luciferase reporter and TK-driven renilla reporter. 48 hours later, firely and renilla luciferase activity was measured. Data points replicate biologic replicates from n=6 independent experiments. Atf3 and Sp7 synergistically activate osteocrin enhancer activity.

Therefore, we tested whether Sp7 and Atf3 may regulate target gene expression in a cooperative manner. Focusing on the -110kB *Ostn* enhancer, we observed that co-expression of Sp7 and Atf3 led to synergistic gains in enhancer reporter activity (**Figure R1**). These interesting results support further study into potential cooperation between Sp7 and Atf3 in promoting osteocyte dendrite development. We believe that this represents a future direction that is clearly beyond the scope of the current manuscript which describes a novel role for Sp7 and its target gene osteocrin in osteoblast-to-osteocyte maturation.

2) Fig 2 & Fig S2: the authors switch back and forth between OCY454 and MC3T3 cells and it is not always clear which cell line was used for which experiment. It appears that they have performed all analyses on both cell lines and obtained consistent results, but please specify this better in the main text rather than in the figure legend.

We thank the reviewer for noting this area for improvement. Indeed, we have performed all analyses on both Ocy454 cells (a conditionally-immortalized murine osteocyte-like cell line) and MC3T3-E1 cells (a well-studied cell line derived from mouse calvariae). The manuscript text and figure legends have been revised throughout to clearly indicate which cell line is used.

3) Suppl figure 1E is missing a panel displaying GFP to show expression of the control vector.

We thank the reviewer for bringing this up. We have performed additional immunoblotting experiments, and in **revised Supplemental Fig. S1d** we now demonstrate strong GFP expression as expected by immunoblotting.

4) What caused the increased intracortical bone formation and increased bone resorption in Sp7OcyKO mice? The authors should test bone turnover markers.

This is an excellent question. As noted in our Discussion, osteocyte apoptosis (which is dramatically increased in the Sp7 mutant mice) has been linked to increased osteocytic RANKL expression, intracortical osteoclastic bone resorption, and subsequent intracortical remodeling. To further test this hypothesis, we performed additional gene expression analysis on cortical bone isolated from control and Sp7 conditional mutant mice. As predicted, Sp7 mutant mice show increased bone RANKL expression (**new Fig. 1h**).

The skeletal phenotype described here is mainly restricted to long bone cortical osteocytes. Indeed, subtle increases in trabecular bone mass are noted (Supplemental Table 1), perhaps as a compensatory change in response to dramatic long bone cortical defects. Since serum bone turnover markers reflect remodeling activity in both cortical and trabecular compartments, we chose not to analyze such markers in this study which focused on histologic and molecular consequences of deleting Sp7 with *Dmp1-Cre* in cortical osteocytes in the axial skeleton.

5) Fig 4G: Could the authors provide the list of 77 identified genes as an actual table embedded in the supplemental figures? At the moment, it is buried within an excel sheet with multiple tabs.

We thank the reviewer for this suggested. We have provided a separate table (**new Supplemental Table 6**) which lists the 77 osteocyte-specific Sp7 target genes identified here.

6) Fig 5: The rescue of the osteocyte phenotype using liver expressed osteocrin is quite impressive. Given the function of osteocrin as a regulator of naturetic peptide signaling, were there effects in bone growth, growth plate, bone mass? Do the authors have a proposed mechanism for this dramatic rescue?

This is a very important question. We did not observe obvious effects on bone length or bone mass in this short term study using AAV8-osteocrin. The main focus of this manuscript is the description of a novel role for the transcription factor Sp7 in osteoblast-to-osteocyte differentiation. In doing so, we identified *Ostn* as an important Sp7 target gene that can rescue osteocyte morphology phenotypes in cells and mice lacking Sp7. Future studies will be needed to completely dissect the downstream signaling mechanisms used by *Ostn* to accomplish these changes in osteocyte morphology. Previous studies have demonstrated that *Ostn* enhances CNP signaling by targeting the CNP clearance receptor NPR3 for degradation in early osteoblast-like cell culture systems (1). Here, we demonstrate that, in mature osteoblasts, the combination of OSTN plus CNP

enhances cGMP signaling (Supplementary Fig. 7a), downstream ERK phosphorylation (**new Supplementary Fig. 7b**), and morphology changes based on high content 'cell painting' imaging (**new Fig. 5m**, **Supplementary Fig. 10 and Supplementary Table 8**). These observations clearly demonstrate potent effects of OSTN signaling on osteocyte cell morphogenesis.

7) Fig S3B: Only one of the three constructs used to knock down *Ostn* expression manages to alter the phalloidin spectrum. Could the authors show the knockdown efficiency of all three constructs to show that this is not due to an off-target effect of the 3rd construct used?

We thank the reviewer for raising this important point. In response to this question, we performed TIDE (Tracking of Indels by Decomposition) analysis (2) to estimate the spectrum and frequency of DNA changes in pools of cells isolated following lentiviral-mediated sgRNA/Cas9 expression and drug selection. In **revised Supplementary Fig. 5b**, we show a clear relationship between genotype (the degree of *Ostn* gene editing) and phenotype (cell morphology as assessed by phalloidin staining and FACS. In addition, robust editing was noted for the other genes tested (*Panx3* and *C1qtnf3*) without morphology changes and changes in phalloiding flow cytometry (**revised Supplementary Fig. 5a-b**).

8) Fig S3D: Please find correlating images to show loss of *Ostn* expression in Sp7 cKO mice. The periosteum does not appear to be shown in the bottom panel.

We thank the reviewer for pointing this out. In **Revised Supplementary Fig. 6**, we now show comparable RNAscope images demonstrating reduced periosteal *Ostn* expression in Sp7 cKO mice versus controls.

9) Fig S4B: If loss of Sp7 alters dendrites and changes phalloidin staining, actin is not a good reference gene to normalize expression levels to.

This is an interesting and important point. We examined *Actb* (β -actin, the housekeeping gene used for RT-qPCR studies) read counts in our bulk RNA-sequencing data (Supplementary Table 7). The absolute abundance of this gene was not affected in any of the cell culture studies (Supplemental Table 7, **Figure R2**). For this reason, we feel confident using *Actb* as our housekeeping gene of choice for RT-qPCR results.

Figure R2. β -actin (*Actb*) read counts in RNA-seq data. No perturbations affected *Actb* mRNA.

10) S4E: Please show *Ostn*-expression in bone, rather than liver tissue.

Cortical bone analysis revealed unchanged *Ostn* expression in AAV8-control versus AAV8-*Ostn* mice (as expected, *Ostn* was lower in Sp7 mutants than controls). This result (**shown in Supplemental Fig. 9**) is consistent with expectations in that AAV8 targets gene expression primarily to liver. Hepatocytes then express high levels of *Ostn* and secrete this protein into the circulation. This model of 'endocrine' osteocrin excess was first established in studies using hepatocyte-specific *Ostn* transgenic mice (3). Similar to this study, here we hypothesize that circulating osteocrin reaches bone to influence osteocyte biology.

11) The human data shown in this manuscript is quite interesting. The authors hypothesize that the OI-related mutation in Sp7 is due to a loss of Sp7-function in osteocytes. If this is true, one would expect to find reduced *Ostn* expression in the human bone samples shown in Fig 7E, while expression of the osteoblast-specific sp7 target genes identified earlier in the paper would not change. Alternatively, could the authors show that overexpression of WT versus mutant Sp7 in vitro does not have an altered ability to induce expression of Ob-specific genes?

This is a very interesting point. Unfortunately, we cannot reliably detect *Ostn* mRNA in human bone samples from control and Sp7-mutant patients. Therefore, we generated *Ocy454* cells expressing human WT and R316C Sp7 and assessed osteocyte- and osteoblast-specific Sp7 target genes. As shown in **new Fig. 7i**, while OI-associated Sp7 expression fails to increase *Ostn* and *Sost* expression, this construct increases osteoblast target genes (*Bglap*, *Alpl*) in a normal matter. These results suggest that the Sp7 R316C mutation selectively interferes with the osteocytic functions of Sp7, a point further addressed in our **expanded discussion**.

Reviewer #2 (Remarks to the Author):

This is exciting work showing the potential role of Sp7 and osteocrin in dendrite formation. The rescue of the Sp7 deletion bone defect by osteocrin is impressive. The most novel aspect of this study is the single cell sorting which identified several clusters including one with high expression of Sost, therefore an osteocyte cluster. Another novel finding is that osteocyte specific genes are targets of Sp7 and distinct from osteoblasts and share similarity with neurons.

We thank the reviewer for these supportive comments.

However, there are some concerns.

It is very hard to prove that a change in osteocyte dendrite formation is responsible for a bone phenotype or disease. It can only be inferred that a defect identified in dendrite formation in *in vitro* experiments translate into the same defect *in vivo*. The bone defect could also be due to other effects of Sp7 such as on matrix formation.

This is an important point. We agree that it can be challenging to ascribe a macroscopic skeletal phenotype completely to a single molecular mechanism. That being said, two lines of evidence support our model that defects in osteocyte dendrites cause increased osteocyte apoptosis and intracortical remodeling in Sp7 conditional knockout mice. First, dendrite defects are seen in non-apoptotic (assessed by TUNEL staining) osteocytes (Fig. 1r, v). Second, osteocrin over-expression, which rescues morphology defects *in vitro* and *in vivo* in Sp7 mutant cells and mice, promotes osteocyte survival and reduces cortical porosity. Nonetheless, we have **revised our Discussion** to highlight the possibilities that other molecular functions of Sp7 in osteocytes may be contributing to the skeletal phenotype seen in these mice.

Fig 1. Increased porosity suggests an increased in osteoclast activity. Greater osteocyte cell death is observed suggesting an increase in Rankl. It is postulated that a reduction in dendrites causes the increase in osteocyte cell death. It would be important to determine if Rankl is elevated in osteocytes in the Sp7 targeted deletion mice.

We thank the reviewer for raising this point, which was also raised by the first reviewer. As discussed above and in the Discussion, *Rankl* (*Tnfrsf11*) mRNA in cortical bone is increased in Sp7 mutant mice (**new Fig. 1h**).

Fig. 2. Normal osteocytes do not show 'spines' on their dendrites. Therefore it is suggested that the dendrites of MC3T3 cells do not represent osteocytes. Do the Ocy454 also show 'spines' on their dendrites? If not, these are a more representative model of primary osteocytes.

We agree that further discussion is needed regarding the spine-like morphology noted in 3D culture of MC3T3 and Ocy454 cells. Notably, Ocy454 cells also show spine-like structures protruding perpendicular to the main axis of their cytoplasmic extensions (**Supplemental Fig. 3**). As noted in the **revised Discussion**, we do not completely understand the significance of these spine-like projections. That being said, it is tempting to speculate that these structures may be similar to 'tethering elements' described at the ultrastructural level used by *bona fide* osteocytes to connect to bone surfaces and amplify mechanical cues (4).

The actual number of dendrites per cell should be counted. The image in 2a is qualitative and not quantitative. For Image 2b, a decrease in phalloidin expression would not necessarily mean a decrease in dendrite number/cell. Again, the actual number of dendrites/cell should be quantitated.

We thank the reviewer for bringing this up. We have quantified osteocyte dendrite number per cell on a blinded basis from all *in vitro* 3D culture experiments, and now show these results in **revised Fig. 2b and 5b**. As expected, these results support our qualitative images and phalloidin flow cytometry data.

It is stated that MC3T3 cells grown in 3D culture have changes consistent with osteocyte maturation (SFig2c and STable 2). What are these changes?

In the revised manuscript, we have significantly expanded the text and discussion. Specifically, in **revised Supplementary Fig. 4a**, we now show a volcano plot with known terminal osteoblast/osteocyte maturation markers (*Dmp1*, *Mmp13*, *Phex*, *Ank*, *Ibsp*) highlighted. In addition, the manuscript text is revised to specifically note well-known osteocyte marker genes whose expression is increased in 3D (versus 2D) culture.

Fig 3. The authors have examined genes regulated by knockdown and over expression of Sp7 in Ocy454 cells. They focus on neuronal genes but ignore the others such as ossification, matrix organization, etc. Just because a pathway is identified does not necessarily mean the pathway is performing the same function in a different cell type, for example, though genes involved in brown fat and inner ear development are also regulated in the Ocy454, that does not mean they have the same function in osteocytes.

This point is well-taken. Multiple previous studies have documented a role for Sp7 in osteoblast lineage commitment and function. The purpose of our work was to investigate novel roles for Sp7 at later stages in the osteoblast/osteocyte lineage. We agree that gene ontology analysis of Sp7 target genes in Ocy454 cells yields multiple terms of potential interest. We chose to focus on neuronal terms for several reasons. First, as highlighted in our introduction, morphologic and physical similar between osteocytic and neuronal network suggests that common molecular pathways may participate in cellular morphology changes in these two cell types. Second, multiple independent Sp7-focused datasets in Ocy454 cells yielded neuronal pathway terms. For example, Sp7 ChIP-seq revealed genes with osteocyte-specific Sp7 enhancer binding which were enriched in terms “motor neuron axon guidance” and “axon extension involved in axon guidance” (Fig. 4c); independently, Sp7 perturbation led to transcriptional changes in groups of genes involved in similar/identical pathways such as “regulation of cell projection organization”, “motor neuron axon guidance”, and “regulation of neuron projection development” (Fig. 3f). Finally, we noted relative expression enrichment of Sp7 target genes and osteocyte-enriched genes in neurons compared to other resident cell types in brain (**Fig. 6o-p**). Taken together, these observations support our focus on neuronal-like GO terms here. We have revised the discussion to further reflect upon comparisons between osteocytes and neurons.

77 direct Sp7 target genes were identified in osteocytes and a significant number of neural genes were expressed. This figure should go into the main text instead of in the supplementary data (Fig S7). This data supports the author’s hypothesis.

We thank the reviewer for this suggestion. We now present our results demonstrating enrichment of Sp7 target genes and osteocyte marker genes in neurons (versus other cell types in brain) as main text (**Fig. 6o-p**).

A clear explanation for why osteocrin was chosen over other Sp7 target genes was not provided. The authors provided no rationale for why they decided to choose this particular target for performing immunostaining of the Sp7 mutant bones. Was this fishing or was there a reason?

We thank the reviewer for raising this important point. We have revised the manuscript text to provide additional clarification regarding why we focused on osteocrin. Specifically, we note the following:

- *Ostn* is expressed in neurons where it regulates dendrite growth (5)
- *Ostn* is a top Sp7-dependent genes identified from our *in vitro* RNA-seq analysis. *Ostn* is significantly downregulated ($\log_2FC = -2.90$) in Sp7 knockout (compared to control) and is significantly upregulated ($\log_2FC = 2.80$) in Sp7 overexpressing Ocy454 cells.
- We searched the *cis*-regulatory regions near the *Ostn* gene, and identified 2 strong Sp7 binding sites based on our Sp7 ChIP-seq data.
- Other than *Ostn*, we also deleted additional genes, including *Panx3* and *C1qtnf3*. Unlike *Ostn*, we did not observe dendrite defects when these genes were deleted.

Fig 6. How was each cluster annotated? Why were the different genes chosen such as *Pdgfr1*, *Mgst3*, etc. What about other markers for different stages of osteoblast to osteocyte differentiation?

We appreciate the opportunity to clarify the methods used for analysis and cluster annotation in our single cell RNA-seq data in Figure 6. For identification of cluster marker genes, we first extracted genes showing relatively restricted expression levels for each cluster compared to the mean of the other cells ($padj < 0.05$). Within this group of genes for each cluster, we then ranked these genes based on descending percentage of

difference (pct.diff) and selected the gene with largest pct.diff as the top marker of that cluster (e.g. *Pdgfrl* for cluster 2). Within each cluster, we then manually analyzed marker genes to check for known osteoblast/osteocyte markers. Through this process, we selected a second marker (e.g. *Bglap3*, *Sost*) for each group of cells. C1 and C2 both have high *Bglap* isoform expression; therefore, these cells were annotated as osteoblast clusters. C6 shows highly enriched *Sost* expression, and C7 has *Cxcl12*-specific expression. Therefore, we annotated C6 as a mature osteocyte cluster and C7 as CAR (CXCL12-abundant reticular) cells. C3 and C5 both show high relative *Enpp1* expression. We annotated these two clusters as differentiating populations given the key role of *Enpp1* in matrix mineralization. *Postn* is identified as one of the top markers in C4. *Postn* is expressed in periosteum and osteocytes during mechanical loading (6). We have included a separate **Supplementary Fig. 13** of feature plots of all clusters (top marker based on pct.diff and second known marker). In addition, **revised Supplementary Fig. 14** now shows individual feature plots for additional marker genes representing different stages of osteocyte maturation.

Supp Fig S6: It would be important to show the expression of additional osteocyte genes such as *Mepe*, *Fgf23*, etc. in addition to the mature osteocyte marker *Sost*, and early osteocyte markers, *Dmp1*, *Pdgn*, and the late osteoblast gene, *Bglap3*. Where is the data showing that *Ptprz1* is a terminal osteocyte marker?

This is an excellent point. As discussed above, we have revised **Supplemental Fig. 14** to include feature plots of additional well-studied osteocyte marker genes (*Mepe*, we could not detect *Fgf23* expression in single-cell data). Previous studies have demonstrated that *Ptprz1* is upregulated during osteoblast-to-osteocyte differentiation (7, 8). Revised **Supplemental Fig. 14** shows a feature plot demonstrating *Ptprz1* expression. In addition, **Supplemental Fig. 18** illustrates concordant changes in expression in the closely-related gene *Ptpr* which is highly restricted to mature osteocytes and nearly absent in the absence of *Sp7*. We have revised the manuscript text accordingly to explain these data which demonstrate that our single cell sequencing results capture important transitions in osteoblast maturation and highlight the role of *Sp7* in this process.

1). With regards to the single cell experiments, why did the authors include calvaria in their bone samples? It is very difficult to remove marrow from calvaria with collagenase digestion/chelation. Could the calvaria bone marrow have skewed the single cell results?

Despite our collagenase/EDTA digestion protocol, we found it necessary to include calvariae in order to obtain sufficient numbers of cells for single cell RNA-sequencing. This sequencing was performed on tdTomato-positive cells isolate from *Dmp1-Cre* ; tdT^{LSL} reporter mice. For this reason, the cells analyzed here all belong to the *Dmp1-Cre* labeled lineage. Rare contaminating hematopoietic cells were excluded as described in the Methods and shown in **Supplemental Fig. 11**.

2). Why are the authors using MC3T3-E1 cells when the *Ocy454* have much higher levels of *Sp7* and there is complete knockdown in the *Ocy454* but not the MC3T3 (supplementary figure S1)? Concerns are expressed above about abnormal dendrites, etc.

We thank the reviewer for providing us an opportunity to clarify this issue. Both *Ocy454* and MC3T3-E1 cells have osteoblastic characteristics before differentiation and recapitulate osteocytic qualities after differentiation (*Ocy454* cells upon switching to 37°C or upon growth in 3D collagen matrix, MC3T3-E1 cells upon growth in 3D collagen matrix). As detailed above, we observe similar and consistent results in *Ocy454* and MC3T3-E1 cells in both systems based on morphology assessment, phalloidin-based flow cytometry, and gene expression profiling. Overall, we believe that our use of two independent cell lines plus mouse and human *in vivo* data represent a strength of this work.

Reviewer #3 (Remarks to the Author):

In this manuscript, Wang et al. identify that osterix/*Sp7* has an unexpected role to regulate osteocyte dendrogenesis, at least in part through osteocrine as a downstream secreted effector. Overall, this work has several notable strengths that result in a high level of enthusiasm and high likely impact. First, this work establishes a new and unexpected function for *Sp7* and takes important steps beyond existing literature towards establishing osteocyte dendrogenesis as a distinct biologic processes required for maintenance of cortical bone strength. This work also correspondingly establishes systems where osteocyte dendrogenesis

can be studied in vivo and in vitro, therefore opening this area to further study. This work also provides an explanation for observations in the literature that several genes likely play fundamentally very distinct roles during the osteoblast to osteocyte transition, than they do during their better studied functions during osteogenic lineage commitment. This report provides an example of how even well studied factors involved with the early stages of osteoblast differentiation can play a distinct dual purpose in later osteocyte transition, by those factors engaging a distinct set of transcriptional partners and by mediating distinct processes, such as osteocyte dendrogenesis, that don't occur during early osteoblast differentiation. Lastly, it is notable that convincing data is presented that the mechanisms identified here operate in humans. Overall, the experiments appear to be technically sound and a large amount of data is provided in support of the conclusions. There are a few important questions, largely about the interpretation of data, that should be resolved.

Major points: 1. This work seeks to draw comparisons between neural dendrogenesis and osteocytic dendrogenesis. Is there a role for osterix in neurons? If feasible, a basic evaluation of *Osx/Sp7* function in neurons would be helpful in judging how universal this *Sp7* pathway is to dendrogenesis. Even if negative, this data would be of interest for clarifying that the convergence between neuronal and osteocytic dendrogenesis happens at the level of a shared set of target gene effectors, despite distinct transcriptional mechanisms being used to engage that dendrogenesis program in each tissue.

This is a very interesting question. *Sp7* expression has been reported during neurogenesis in the olfactory bulb, cortex, and cerebellum. To date, functional studies have only focused on the olfactory bulb, where no obvious defects are noted in germline *Sp7*-null mice (9, 10). While studies on *Sp7* during neurogenesis fall somewhat outside the scope of the current manuscript, we did examine CNS tissue from *Sp7-Cre ; tdTomato^{LSL}* mice where, consistent with previous publications, we also observe subpopulations of neurons clearly marked with *tdTomato* expression (new Supplemental Fig. 19). We have revised the Discussion to address this question, and to note that additional studies are needed to explore whether *Sp7* plays a functional role in neurons. The clear enrichment in neurons (versus other cell types in brain) of *Sp7* target genes and osteocyte-enriched transcripts (Fig. 6o-p) further supports the notion that common molecular programs exist between osteocytes and neurons.

2. Some clarification and additional data is needed regarding the uCT studies. What methods were used to measure cortical porosity? Specifically, are the values reported in Fig 1e percent of pore volume over total cortical volume or is some other volume used as the denominator? Is trabecular bone mass intact in these mice? Is there a calvarial phenotype? The calvarial phenotype is of interest given the use of the calvarial site for some of the studies characterizing dendritic morphology. Is the overall cortical diameter unchanged vs controls? Have these mice been observed to display spontaneous fractures?

We appreciate the opportunity to clarify these microCT data. **First**, cortical porosity was measured as the total area of pores in the cortex divided by the total area of the cortex [cortical porosity (%) = (pore area / cortex area) x 100]. We have revised the methods section to include this information. **Second**, trabecular parameters (metaphyseal bone in the distal femur) are included in Supplement Table 1a. In the presence of a dramatic cortical bone phenotype, there is a mild increase in trabecular bone mass. We believe that this change is compensatory in nature since *Dmp1-Cre* deletion is less efficient in osteocytes embedded in trabecular bone. **Third**, the reviewer's questions about calvarial bone are quite interesting. Our analysis of osteocyte morphology was largely restricted to cortical bone in the axial skeleton. However, we did use THG imaging to assess osteocyte morphology in calvariae, where dendrite defects were also noted in *Sp7* mutant mice (Fig. 1j). We scanned skulls from 6 week old control and *Sp7* mutant mice. As expected, *Sp7* mutant calvariae show normal overall morphology but areas of hypomineralization (Figure R3). **Fourth**, the questions about cortical morphology are quite important. We have now measured mid-shaft cortical diameters (*DMin* and *DMax*) and report these parameters in

Figure R3. Representative skull microCT from 6 week old control and *Sp7* mutant mice.

revised Supplemental Table 1a. Consistent with the dramatic cortical phenotype, compensatory changes in cortical diameter are present. The **results text has been revised** to include explicit statements about changes in trabecular bone volume fractions and cortical diameter in the Sp7 mutants. **Finally**, the question about spontaneous fractures is quite interesting. In this study, we sacrificed all Sp7 mutant mice at a relatively young age (6-8 weeks old) for analysis of cortical bone histology and microCT. In the course of these studies, we did observe a few (2-3) spontaneous long bone fractures in mutant (but not control) mice. Future studies are needed to assess fracture incidence over longer periods of time in a dedicated cohort of animals.

Minor points

1. Generally the methodologic description of FACS and single cell sequencing analysis generally provides a good degree of detail, but a few additional details would both help in judging this data and also facilitate reproduction. The FACS plots used for DAPI sorting and doublet exclusion should be provided in supplemental figures. Was filtering based on relative amounts of ribosomal or mitochondrial transcripts performed?

We thank the reviewer for raising these important questions about our methods. We have **revised Supplemental Fig. 11a** to include FACS plots to demonstrate methods for doublet exclusion and sorting parameters. Single cell profiling datasets were filtered based on relative amounts of mitochondrial transcripts using published methods (11). 10X Genomics also has a technical note on filtering datasets (https://assets.ctfassets.net/an68im79xiti/4tVumiyINGgAeoCg8SiWGG/1cf0888200d668142612c8d3f3679cf4/CG000130_10x_Technical_Note_DeadCell_Removal_RevA.pdf). Based on this information, we filtered on cells with less than 10% mitochondrial reads. **Supplemental Fig. 11b** shows quality control metrics for mitochondrial transcripts.

While osteocytes are generally considered post-mitotic, it is possible that some of the less mature populations captured could be entering cell cycle. Do any of the clusters show strong differences in cell cycle indicating that this is driving the clustering?

This is an important point. We used pre-defined S-phase and G-to-M transition gene lists in Seurat (https://satijalab.org/seurat/archive/v3.1/cell_cycle_vignette.html) to assess cell cycle signatures across our cell clusters. As expected, there is a high concentration of these two scores in the Mki67+ cluster (cluster 8) with essentially no mitotic gene expression in the other clusters (**Figure R4**).

A similar question pertains to whether there are any clusters that show clear differences in the number of UMIs per cell equivalent, though it is appreciated that likely doublet filtering and some normalization steps were applied.

This is also a very important point. We filtered out on doublets based on whether there is a co-occurrence of marker genes that might suggest as much (osteoblast/osteocyte co-localization). **Figure R5** shows plots based on nUMI and nGene. A gradient of the number of genes/UMIs per cell exists which may reflect intrinsic biological difference between these clusters. For example, there are clearly higher metrics in a portion of the dividing populations, which make

Figure R4. S phase (left) and G2M (right) genes across clusters. As expected, only cluster 8 (defined as Mki67 high) cells show enrichment of cell cycle genes.

Figure R5. Gene number (left) and UMI number (right) across clusters.

sense as these cells are actively doubling themselves.

2. If feasible, in Fig 1g, if the morphologic parameters of the dendrites can be explored in further detail, this would enhance the data presentation. Can mean total dendrite length be distinguished and reported separately from the mean number of branch points each dendrite displays (perhaps with a normalization for dendrite length) using the harmonic imaging data? Is this a phenotype of mean dendrite length or a defect in branching or both? It would appear feasible to count the branch points manually if needed. As this area of research becomes more sophisticated, it seems likely that there will be phenotypes that separately influence dendrite extension and branching, so distinguishing these properties is of interest. Along similar lines, can osteocyte-osteocyte dendritic connections be counted in the harmonic imaging data, this would also be of interest to emphasize that not only is dendritic length reduced, but that this results in loss of functional contacts among osteocytes.

We thank the reviewer for this very interesting series of thoughtful questions. We explored the morphological parameters mentioned above in our THG datasets where the number of osteocyte-osteocyte dendritic connections was quantified manually using ImageJ software. The mean number of osteocyte-osteocyte dendritic connections per osteocyte was found to be 5.10 ± 0.21 for wild-type and 3.97 ± 0.29 for $Sp7^{OcyKO}$. The error given for all the values stated is the standard deviation of the mean values for 2 mice. These results (shown in **new Fig. 1o**) indicate a statistically significant difference ($p < 0.05$) using a two-tailed t-test.

In addition, we re-analyzed the data using microscopy image analysis software (Imaris) to obtain the average dendrite length, which represents the average length of individual dendrite sections between branch points, as well as the average number of branch points were counted and normalized to the number of osteocytes in the volume. Quantitative analysis with Imaris was performed using the Filament Tracer tool, where filaments were traced and visualized as shown in Fig. 1j. Specifically, the logarithm of the 3D stacks of cropped THG intensity images was loaded into the Filament Tracer tool, and the following parameters were used: Dendrite Starting Point Diameter = $6.000 \mu\text{m}$, Dendrite Seed Point Diameter = $0.882 \mu\text{m}$, Dendrite Starting Point Threshold Low = Automatic, Dendrite Starting Point Threshold High = Automatic, Dendrite Seed Point Threshold = Automatic, Diameter of Seed Point Remove = true, Dendrite Diameter Threshold = 15.616 , Dendrite Diameter Algorithm = Distance Map and Spine Detect = None. The parameters were determined by observing which settings trace out the dendrites most appropriately, under the consultation of technical assistance at Imaris. The mean dendrite length was found to be $6.0 \pm 0.1 \mu\text{m}$ for wild-type and $6.3 \pm 0.3 \mu\text{m}$ for $Sp7^{OcyKO}$. The mean number of dendrite branch points per osteocyte was found to be 290 ± 120 branch points for wild-type and 217 ± 36 branch points for $Sp7^{OcyKO}$. The error given for all the values stated is the standard deviation of the mean values for 2 mice. These results show a trend of decrease but do not indicate a statistically significant difference between the mean dendrite length and the mean number of dendrite branch points per osteocyte between wild-type and $Sp7^{OcyKO}$ mice. It should also be noted that the resolution of the THG microscope ($\sim 0.5 \mu\text{m}$ lateral and ~ 3.0 axial) makes it difficult to ascertain small differences in dendrite length between wild-type and $Sp7^{OcyKO}$. Future studies, where a larger cohort of mice are imaged using THG, are needed to better define the effects of Sp7 deletion on these 'next generation' osteocyte morphology parameters. We feel that such investigation falls outside the scope of the current work whose focus is on the molecular mechanism used by Sp7 to regulate osteocyte morphology.

3. Similar to the comment above, it would strengthen the presentation if density of both lacunae and osteocytes per areal unit of bone cortex can be reported. Fig 1 makes a convincing case the osteocytes die at an increased rate in the $Sp7^{dmp1-cre}$ mice, resulting in a large number of empty lacunae. It then becomes of interest to understand whether the rate of osteocyte generation/the rate of osteoblasts proceeding to become matrix embedded osteocytes is impacted. Perhaps measuring the density of lacunae and then normalizing to the bone formation rate measured by histomorphometry would allow estimation of the rate of osteocyte generation? Clarity whether the rate of osteocyte generation is impacted will be useful to aid in interpreting the single cell seq data in Fig 6 in terms of understanding whether there is clear evidence for an osteoblast to osteocyte differentiation block in vivo.

This is also an interesting series of questions. Of note, we measured osteocyte parameters (lacunar density, apoptosis, dendrites) in cortical bone, as reported in Fig. 1s-t. We did perform dynamic histomorphometry based upon dual calcein/demeclocycline labeling to measure the bone formation rate. However, this

quantification was performed on trabecular bone surfaces. We did not dramatic qualitative increases in intracortical bone formation in Sp7 mutant mice (Fig. 1c). However, routine measurement of the endocortical or periosteal bone formation rate is extremely challenging on sagittal sections in wild type mice and was not performed here. This information would be needed to address this interesting suggestion. Alternative methods to assess osteoblast-to-osteocyte differentiation *in vivo* may be preferred. For example, one could imagine a lineage tracing approach that would selectively label osteoblasts (but not osteocytes) which could then be used to track their maturation into osteocytes. However, all of the currently-available osteoblast-specific Cre^{ER12} transgenic and knock-in mouse lines also show some activity in osteocytes, thus precluding such analysis. Therefore, a focus of future studies to build upon the current work will be to develop novel methods to assess osteoblast-to-osteocyte differentiation *in situ*.

4. As an optional suggestion, it may be of interest to highlight a few of the Sp7 dependent genes that are known to be or are plausibly involved in dendrogenesis. These could be distilled into a small figure or supplementary figure perhaps showing an expression heatmap with a few manually selected with the degree of change upon Sp7 loss and gain of function. This would also provide an opportunity to briefly highlight or discuss some relevant neural literature that establishes these genes as involved with neuronal dendrogenesis. This would help to further develop the claim that Sp7 is a regulator of dendrogenesis and will also likely be of interest to most readers.

We appreciate this suggestion. **Fig. 3f has been revised** to highlight genes of interest in each GO group, including genes in the “neuron axon guidance” term whose expression is regulated by Sp7. **We have also revised the text/references to highlight connections between these Sp7-regulated genes in bone cells and neuronal dendrogenesis.**

5. *Ostn* is described as an osteocyte-specific transcript, but the data in Fig 3c suggests that it is expressed in what appear to be periosteal osteoblasts based on morphology. Clarification or, if needed, changes to terminology used to describe *Ostn* expressing cells would be helpful. If *Ostn* is expressed in osteoblasts, does this indicate that the cells receptive to *ostn* signals are later stage cells where dendrogenesis occurs and that *ostn* is not acting in an autocrine manner? Was *Ostn* expressed in the scRNA-seq data and, if so, which clusters displayed expression? Was this expression diminished in Sp7-deficient samples by scRNA-seq? It seems like there is likely a complex effect of Sp7 such that it has distinct impacts at multiple distinct steps in the osteocyte differentiation process. Exploring the existing data further from this perspective to start to make some of these issues more explicit in the manuscript would likely aid with clarifying these effects.

We again appreciate this thoughtful and interesting series of questions. Due to limitations in sensitivity in our 3' single cell RNA-seq (10X Genomics platform) profiling, we cannot detect *Ostn* expression in these libraries from control or Sp7 mutant mice. Future studies with more sensitive single cell sequencing methods (such as SmartSeq2) are likely needed to detect endogenous *Ostn* expressing at this level. Based on our *in situ* hybridization studies (**Supplemental Fig. 6**, *Ostn* is expressed in periosteal cells and, to a lower extent, in early embedding osteocytes. This *Ostn* expression profile is consistent with what others have reported in mice and humans (12-14). We have revised the manuscript text to reflect the *Ostn* expression profile in bone. Local expression of this secreted factor likely acts in an autocrine or paracrine manner to promote dendrite formation during osteocyte maturation.

6. Related to the above point, the *Ostn* rescue is very impressive and has interesting potential therapeutic implications. However, there appears to be a disconnect between this *Ostn* rescue with the idea that Sp7 is directly promoting transcription of effector genes for dendrogenesis. Is it the case that the main Sp7-dependent effector is *Ostn* and the various dendrogenesis effector genes are only indirectly regulated by Sp7 via an *Ostn*-dependent autocrine loop? Or is it instead the case that both Sp7 and *Ostn* have additive and direct effects on the transcription of dendrogenesis effector genes? It is appreciated that it can be challenging to fully disentangle these possibilities. However, perhaps the Sp7 ChIP data combined with the various transcriptional profiling studies already performed can be used to at least partially clarify this question.

We whole-heartedly agree that it can be challenging to precisely define direct versus indirect effects of Sp7 and *Ostn* at the level of osteocyte dendrite formation and maintenance. We appreciate the excellent suggestion to merge our Sp7 ChIP-seq results with various transcriptomic datasets. We focused analysis on

1042 genes closest to the top 1000 (by read density) osteocyte-specific distal enhancer Sp7 binding sites. Next, we cross-referenced this group of 1042 genes with the list of differentially-expressed genes regulated by Sp7 knockdown (2873 genes), Sp7 over-expression (493 genes), or Osn over-expression (3851 genes). Of the 2873 genes regulated by Sp7 knockdown, 282 (9.82%) show high density Sp7 ChIP-seq peaks. Of the 493 genes regulated by Sp7 over-expression, 50 (10.14%) show high density Sp7 ChIP-seq peaks. Finally, of the 3851 genes regulated by Osn over-expression, 385 (10%) show high density Sp7 enhancer peaks. Similar enrichment patterns between Osn- and Sp7-regulated genes and Sp7 binding sites are seen when binding sites are restricted to genes with neuronal gene ontology terms. Taken together, this analysis demonstrates similar enrichment of Sp7 enhancer peaks in genes regulated by Sp7 or Osn. Therefore, it is possible that Sp7 both regulates Osn expression (as we show here) **and** participates in gene regulation in response to Osn signaling. That being said, providing excess Osn to Sp7-deficient cells and mice leads to a clear rescue phenotype, suggesting that Sp7 may function preferentially upstream (rather than downstream) of Osn. Finally, it is worth noting here that Sp7 also binds to and controls expression of other genes with potential roles of osteocyte dendrite formation. Our studies here with high dose osteocrin over-expression suggest an important functional role for Osn, but it remains possible that other Sp7 target also play critical roles during normal osteocyte maturation. The **manuscript discussion has been revised** to address the important points raised by this series of questions.

7. The finding of CAR cells in the DMP1-cre scRNA-seq studies is unexpected, as these cells are not normally near the cortical surface and are thus not often considered in relationship to osteocytes. This should likely be commented on in the manuscript.

We agree, and also were surprised to note CXCL12 abundant reticular cells among those labeled with Dmp1-Cre. However, others have also reported that Dmp1-Cre labels CAR cells (15). We have included this reference and revised the manuscript to highlight this observation. Our scRNA-seq results clearly highlight the fact that many Cre lines label a somewhat heterogeneous mixture of cells.

8. Can any of the Sp7 regulated candidate effectors of dendrogenesis be detected in the scRNA-seq data? Does this provide insight into which cluster serves as the point where dendrogenesis is initiated? It would not be surprising if dendrogenesis is restricted to SOST+ classical mature osteocytes, but would be interesting if it is primed at an earlier stage of differentiation. Can differences in expression of these dendrogenesis effector genes also be appreciated in scRNA-seq studies of Sp7-deficient samples? This type of analysis may also be helpful for better understanding the relationship among the several processes occurring that are regulated by Sp7: 1) regulation of osn expression, possibly in an osteoblast-like cell, 2) regulation of dendrogenesis, possibly through both osn-dependent and osn-independent by sp7-dependent mechanisms, 3) a differentiation blockade as suggested by the single cell seq clustering data, 4) osteocytic cell death in the absence of Sp7.

This is an extremely thoughtful series of questions and excellent suggestions. Our Ocy454 cell RNA-seq and ChIP-seq datasets identified a group of neuronal-like genes whose expression is directly regulated in osteocytes. In response to this question, we assessed the expression of these genes in our scRNA-seq libraries from control and Sp7 mutant mice. As noted above, sensitivity of 3' scRNA-seq using the 10X Genomics platform is limited. Therefore, we were only able to confidently detect expression of a subset of these genes. Representative feature plots of such detectable genes are shown in **new Supplemental Fig. 18** and described in the revised manuscript text. In general, two such categories of 'neuronal' Sp7 target genes emerge from this analysis. One group of genes (exemplified by *Kank1* and *Cryab*) is expressed in osteocytes (cluster 6) and osteoblasts (clusters 1 and 2) in control mice, and down-regulated in these clusters in the absence of Sp7 with concomitant induction in 'intermediate' cells in clusters 3 and 5. A second group of genes (exemplified by *Ackr3*, *Ptpr*, and *Sost*) is restricted to osteocytes (cluster 6) in control mice, and down-regulated in cluster 6 with subtle up-regulation in 'intermediate' cluster 5 in Sp7 mutant animals. Overall, these results suggest the process of dendrite formation is complex and involves induction of distinct waves of Sp7-dependent genes.

9. It appears that the demographic information for the patients contributing specimens is in reference 15? If available, it may be helpful to reproduce some of this information in the methods or corresponding figure legend to better clarify the characteristics of patients contributing specimens to the study.

We thank the reviewer for this important comment. We have **revised the manuscript (methods)** to include appropriate demographic information for the patients contributing specimens to the study.

Sincerely,

Marc Wein, MD, PhD

References

1. Moffatt P, Thomas G, Sellin K, Bessette MC, Lafreniere F, Akhouayri O, et al. Osteocrin is a specific ligand of the natriuretic Peptide clearance receptor that modulates bone growth. *J Biol Chem* . 2007;282(50):36454-62.
2. Brinkman EK, Chen T, Amendola M, and van Steensel B. Easy quantitative assessment of genome editing by sequence trace decomposition. *Nucleic Acids Res*. 2014;42(22):e168.
3. Kanai Y, Yasoda A, Mori KP, Watanabe-Takano H, Nagai-Okatani C, Yamashita Y, et al. Circulating osteocrin stimulates bone growth by limiting C-type natriuretic peptide clearance. *J Clin Invest*. 2017;127(11):4136-47.
4. Han Y, Cowin SC, Schaffler MB, and Weinbaum S. Mechanotransduction and strain amplification in osteocyte cell processes. *Proceedings of the National Academy of Sciences of the United States of America*. 2004;101(47):16689-94.
5. Ataman B, Boulting GL, Harmin DA, Yang MG, Baker-Salisbury M, Yap EL, et al. Evolution of Osteocrin as an activity-regulated factor in the primate brain. *Nature*. 2016;539(7628):242-7.
6. Gerbaix M, Vico L, Ferrari SL, and Bonnet N. Periostin expression contributes to cortical bone loss during unloading. *Bone*. 2015;71:94-100.
7. Schinke T, Gebauer M, Schilling AF, Lamprianou S, Priemel M, Mueledner C, et al. The protein tyrosine phosphatase Rptpzeta is expressed in differentiated osteoblasts and affects bone formation in mice. *Bone*. 2008;42(3):524-34.
8. Paic F, Igwe JC, Nori R, Kronenberg MS, Franceschetti T, Harrington P, et al. Identification of differentially expressed genes between osteoblasts and osteocytes. *Bone*. 2009;45(4):682-92.
9. Park JS, Baek WY, Kim YH, and Kim JE. In vivo expression of Osterix in mature granule cells of adult mouse olfactory bulb. *Biochemical and biophysical research communications*. 2011;407(4):842-7.
10. Park JS, Park GI, and Kim JE. Osterix is dispensable for the development of the mouse olfactory bulb. *Biochemical and biophysical research communications*. 2016;478(1):110-5.
11. Ilicic T, Kim JK, Kolodziejczyk AA, Bagger FO, McCarthy DJ, Marioni JC, et al. Classification of low quality cells from single-cell RNA-seq data. *Genome biology*. 2016;17:29.
12. Thomas G, Moffatt P, Salois P, Gaumont MH, Gingras R, Godin E, et al. Osteocrin, a novel bone-specific secreted protein that modulates the osteoblast phenotype. *J Biol Chem* . 2003;278(50):50563-71.
13. Bord S, Ireland DC, Moffatt P, Thomas GP, and Compston JE. Characterization of osteocrin expression in human bone. *J Histochem Cytochem*. 2005;53(10):1181-7.
14. Watanabe-Takano H, Ochi H, Chiba A, Matsuo A, Kanai Y, Fukuhara S, et al. Mechanical load regulates bone growth via periosteal Osteocrin. *Cell reports*. 2021;36(2):109380.
15. Zhang J, and Link DC. Targeting of Mesenchymal Stromal Cells by Cre-Recombinase Transgenes Commonly Used to Target Osteoblast Lineage Cells. *Journal of bone and mineral research : the official journal of the American Society for Bone and Mineral Research*. 2016.

Reviewers' Comments:

Reviewer #1:

Remarks to the Author:

The authors have nicely addressed or clarified my comments. I agree that some of the deeper mechanistic questions can be addressed in future work. The reorganization of the manuscript makes it easier to follow the very significant amount of work performed by the group.

Reviewer #2:

Remarks to the Author:

The manuscript is much improved.

As pdpn was decreased in cluster 6 in the KO, you might want to cite previous references on the role of this protein in osteocyte dendrite elongation (Zhang et al, Mol Cell Biol 2006).

Fig S2: the punctate structures could be vesicles or exosomes as described by Sarah Dallas.

Fig S7: western blot should be quantitated.

Fig S9: need description of (b) in figure legend.

Fig S10 (a) Is this missing MC3T3 WT?

Reviewer #3:

Remarks to the Author:

The authors have provided a extremely thorough and thoughtful response to the questions raised in the initial round of review. In particular, questions about re-analysis of several datasets generated have been given careful consideration. All questions have been addressed. This manuscript will generate significant interest for providing one the first considerations for how osteocyte dendrogenesis impacts overall bone physiology and for uncovering unexpected functions for both Sp7 and Osn.

We thank the reviewers for their supportive comments and interest in our work. At this point, only reviewer #2 had outstanding question and suggestions. Our responses to these helpful points are shown below in blue text.

As pdpn was decreased in cluster 6 in the KO, you might want to cite previous references on the role of this protein in osteocyte dendrite elongation (Zhang et al, Mol Cell Biol 2006).

This is an excellent point. The corresponding manuscript text has been revised to include this seminal reference to the role of Pdpn/E11 in osteocyte dendrite elongation.

Fig S2: the punctate structures could be vesicles or exosomes as described by Sarah Dallas.

This is a very interesting point. The corresponding supplemental figure legend has been revised, and reference to a recent review by Sarah Dallas on this topic has been included.

Fig S7: western blot should be quantitated.

We have quantified the immunoblots in Figure S7 and new Figures S7c and d show this quantification. As expected, combination treatment of OSTN plus CNP increases ERK phosphorylation.

Fig S9: need description of (b) in figure legend.

The legend for Figure S9 has been revised to clearly explain that panel (b) shows gene expression in femur.

Fig S10 (a) Is this missing MC3T3 WT?

WT MC3T3 cell correlation matrices are not shown in Figure S10a because these results are shown in main text figure 5m.